# Improving Generalization of Dynamic Graph Learning via Environment Prompt

Kuo Yang[1,2], Zhengyang Zhou[1,2,3*], Qihe Huang[1,2], Limin Li[1,2],
Yuxuan Liang[4], Yang Wang[1,2*]

[1] University of Science and Technology of China (USTC), Hefei, China
[2] Suzhou Institute for Advanced Research, USTC, Suzhou, China
[3] State Key Laboratory of Resources and Environmental Information System, Beijing, China
[4] The Hong Kong University of Science and Technology (Guangzhou), Guangzhou, China
yangkuo@mail.ustc.edu.cn, zzy0929@ustc.edu.cn, {hqh, lilimin}@mail.ustc.edu.cn,
yuxliang@outlook.com, angyan@ustc.edu.cn

## Abstract

Out-of-distribution (OOD) generalization issue is a well-known challenge within
deep learning tasks. In dynamic graphs, the change of temporal environments is
regarded as the main cause of data distribution shift. While numerous OOD stud-
ies focusing on environment factors have achieved remarkable performance, they
still fail to systematically solve the two issue of environment inference and utiliza-
tion. In this work, we propose a novel dynamic graph learning model named EpoD
based on prompt learning and structural causal model to comprehensively enhance
both environment inference and utilization. Inspired by the superior performance
of prompt learning in understanding underlying semantic and causal associations,
we first design a self-prompted learning mechanism to infer unseen environment
factors. We then rethink the role of environment variable within spatio-temporal
causal structure model, and introduce a novel causal pathway where dynamic sub-
graphs serve as mediating variables. The extracted dynamic subgraph can effec-
tively capture the data distribution shift by incorporating the inferred environment
variables into the node-wise dependencies. Theoretical discussions and intuitive
analysis support the generalizability and interpretability of EpoD. Extensive exper-
iments on seven real-world datasets across domains showcase the superiority of
EpoD against baselines, and toy example experiments further verify the powerful
interpretability and rationality of our EpoD.

## 1 Introduction

Dynamic graph learning aims to capture the evolution patterns of individual feature and global topol-
ogy in spatio-temporal graphs over time. It has extensive applications in real-world scenarios, such
as social relationship analysis [6, 61], traffic flow forecasting [3, 59, 64] and air quality prediction
[16, 23]. The dynamic evolution is a prominent characteristic of spatio-temporal graphs [4, 16, 47],
such as human interest and social development naturally undergo changes over time. This charac-
teristic inevitably gives rise to a issue of data distribution shifts. Given this issue, enabling models
to get temporal out-of-distribution (OOD) generalization ability poses a major challenge in dynamic
graph learning [51, 55, 63].

Actually, recent studies have paid attention to tackling the issue of temporal OOD generalization
[53, 58, 61]. They demonstrate that unseen temporal environments contribute to such distribution

---

*Yang Wang and Zhengyang Zhou are corresponding authors.

38th Conference on Neural Information Processing Systems (NeurIPS 2024).

shift. For example, potential urban-hosted events can lead to out-of-distribution traffic state, and unrecorded academic communication between individuals in citation networks may hide new patterns of cooperation. Therefore, the research line relying on environment inference present a promising solution for addressing the temporal OOD issue. This paradigm focuses on inferring underlying environment factors, and utilizing the extracted environment information to enhance the robustness of dynamic graph learning against environment shifts. Although some works have achieved impressive performance [53, 58], there remain limitations in both environment inference and utilization.

The judgment of existing limitations stems from two crucial observations. **Firstly**, unseen environments invariably encompass a wide range of factors, posing challenges in accurately determining their quantities and scales. However, existing methods often rely on a predefined scale environment codebook for inferring unseen environments [53, 58], which may infer unrealistic environment results. **Secondly**, the shift of environments in dynamic graphs fundamentally reflects in the changes of structural associations [48, 65]. A real-world example is that the change of weather alters the future flow of the traffic network by changing human's trajectory. Nevertheless, existing methods often prioritize using the inferred environment as additional information to augment raw features, overlooking capturing the evolving structural associations [53, 58, 61]. Therefore, the existing dynamic graph OOD efforts face issues in both environment extraction and utilization, and the comprehensive solution to address both problems is currently lacking.

To tackle these limitations, we propose an **E**nvironment-**pro**mpted **D**ynamic Graph Learning (EpoD) architecture. **Firstly**, we propose a novel self-prompted learning mechanism to infer underlying environment representation. Given the lacking of environment labels and explicit scaling of environments, our aim is to guide the network generalizing environment factors from historical data in autonomous manner. The practices of language model inference on underlying semantics inspires us to utilize prompt learning to achieve this goal [14, 31]. We propose a self-prompted learning mechanism for spatio-temporal data to infer environment variables from historical data. By designing learnable prompt tokens and an interactive prompt-answer squeezing mechanism, we enable the model to effectively infer the compact and informative environment representations. **Secondly**, we propose a novel Structural Causal Model (SCM) with dynamic subgraph as mediating variable to enhance the adaptability of the network to environment shifts. Different from some approaches that obtain the causal subgraph by partitioning the original graph [61], we design a node-centered subgraph extractor specifically tailored for spatio-temporal data. This design is derived from a profound understanding of dynamic graph that the shift of environments within dynamic graphs invariably result in the changes of these asymmetric correlations between nodes. Our node-centered dynamic subgraphs extractor can capture node-wise asymmetry, where each node has its unique subgraph based on its environment states. **Lastly**, we conduct comprehensive experiments to evaluate the generalizability of EpoD. On the one hand, we perform experiments over multiple cross-domain datasets and introduce a more intricate long-series prediction task. On the other hand, we design an environment-shaded toy dataset, named *EnvST*, to verify the generalization ability of EpoD. The results show that EpoD can precisely perceive environment factors, and the generated dynamic subgraphs are equipped with both generalizability and interpretability. Our **contributions** are summarized as follows:

- We systematically investigate the environment-based efforts to tackle the temporal OOD issue, and observe the limitations of existing works in environment inference and utilization.

- To address existing challenges, we propose a novel **E**nvironment-**pro**mpted **D**ynamic Graph Learning (EpoD) architecture. Specifically, we introduce a self-prompted learning mechanism for spatio-temporal data to infer underlying environment variables without preset scale. For the exploitation of inferred environment factors, we propose a structural causal model with dynamic subgraphs for mediating variables to capture the effect of environment variable shifts on the data distribution. Our work presents a pioneering practice jointly focusing and solving the issue of environment inference and utilization.

- We conduct experiments over multiple cross-domain datasets, including traffics and social networks, to verify the effectiveness of our framework. And a toy dataset is designed to demonstrate the generalizability and interpretability of EpoD.

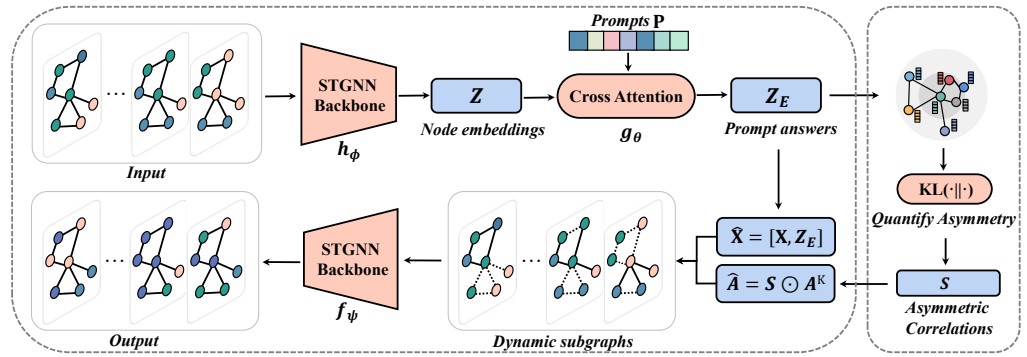

Figure 1: The architecture of EpoD. Left panel: the prediction of future evolution based on historical observations. Right panel: the extraction process of node-centered dynamic subgraph.

## 2   Background

**Preliminaries.** We denote $\mathcal{G} = \{\mathcal{G}^t\}_{t=1}^{T}$ as a dynamic graph across $T$ steps, where $\mathcal{G}^t = (\mathcal{V}^t, \mathcal{X}^t, \mathcal{A}^t)$ represents a snapshot of the graph at step $t$. The tensor $\mathcal{X}^t \in \mathbb{R}^{N \times D}$ indicates observed features of $N = |\mathcal{V}^t|$ nodes at step $t$, where $D$ denotes the feature dimension. $\mathcal{A}^t \in \{0,1\}^{N \times N}$ is the adjacency matrix describing the connectivity of graph $\mathcal{G}^t$. Given the historical data $\mathbf{X} = \{\mathcal{G}^t\}_{t=1}^{T}$, dynamic graph learning aims to predict the future evolution patterns $\mathbf{Y} = \{\mathcal{G}^t\}_{t=T+1}^{T+K}$, where $K$ denotes the number of predicted future time steps. Historical observations $\mathbf{X}$ can be divided into the accessible environment features $\mathbf{X_X}$ and observed labels with evolution patterns $\mathbf{Y_X}$, e.g., volumes in traffic datasets, or links in social networks.

**Problem Definition.** There is a consensus that unseen environment factor, $\mathbf{E}$, considered as the primary reason for temporal OOD issue [58, 53]. In this work, we focus on capturing the invariant evolution pattern of dynamic graph to tackle the temporal OOD issue.

**Spatio-Temporal Graph Forecasting for OOD issue.** Our work aims to systematically tackle the challenges in existing environment-centered OOD approaches. Our specific practice in environment inference and utilization is significantly distinct from existing works and has substantial improvements. *On the one hand*, the self-prompted environment inference framework aims to guide the network to adaptively infer environment variables from historical data by using well-designed prompt tokens. Different from existing efforts relying on predefined environment scale [53, 58], our methods advantageously eliminates human biases and ensures accurate extraction of the environment from real historical data. *On the other hand*, we propose a novel SCM with dynamic subgraph as mediating variable. Compared to simply attaching environment embeddings to existing representations, our approach is more consistent with the understanding of dynamic graph data from a causal perspective. Furthermore, unlike some approaches that rely on partition strategies commonly used in static graph learning to extract subgraphs [61], we propose a node-centered dynamic subgraph extraction method that is better suited for spatio-temporal graph scenarios.

**Prompt Learning.** Prompt learning was initially introduced to address the challenge of data scarcity in language models [7, 12, 11]. By utilizing well-designed prompts, the model can effectively capture a broader space of data distributions and patterns during training, which is better at adapting to input samples from different distributions. For an extended period, the complexity of prompt design has been a hindrance preventing prompt learning from achieving broader applications. In contrast to language data, human beings lack the intuitive cognition of both image and graph data, making it challenging to design interpretable prompts templates. Recently, learnable prompts have also been proven to have superior performance [29, 20]. Therefore, prompt learning is thriving in computer vision research and graph learning [22, 30, 39]. Recently, the efforts of using prompt learning to enhance spatio-temporal prediction is beginning to emerge, including [62] and [60]. The former focuses on multi-attribute forecasting; the latter aims to enhance model generalization ability. However, there is lacking systematical research on prompt learning addressing temporal OOD problems.

# 3 Methodology

In this section, we introduce a novel **E**nvironment-**pro**mpted **D**ynamic Graph Learning architecture named EpoD. First, we propose a self-prompted learning mechanism to realize the awareness of unseen environment factors. Second, we revisit the winding causal path from environment to graph evolution, and a spatio-temporal learning framework utilizing dynamic subgraph as mediating variables is presented. Last, we provide theoretical analysis of EpoD from causal perspective to interpret its excellent generalization ability.

## 3.1 Self-prompted learning for environment awareness

Existing methods typically infer environments by predefining the scale of the environment codebook, which introduces human bias and has the potential to cause performance degradation. To tackle this issue, we introduce an environment inference principle that extract underlying environment representations from historical observations without predefining scales. Inspired by the remarkable success of prompt learning in inferring underlying semantics and the multimodal generalization of large language models (LLMs) [49, 35, 39], we propose a self-prompted learning mechanism (SPL) to realize this environment inference principle.

**Spatial-specified prompt design.** Our SPL focuses on guiding models to effectively extract environment variables from observed data by well-designed prompts. Therefore, the initial challenge we need to address is the design of prompt tokens. Within spatio-temporal graphs, it is common for different nodes to exhibit diverse evolution patterns. Thus, the design of prompts should be specified on spatial aspect to reflect node-wise distinctive evolution state. Given a dynamic graph $\mathcal{G}$, we assign a prompt token $\mathbf{p}_i$ for each node, which is shared across temporal steps. However, the absence of prior knowledge about environment information hinders us from adopting the template-based approach to initialize $\mathbf{p}_i$. Fortunately, learnable prompts have revealed satisfactory performances on capturing hidden mappings [39, 30]. Therefore, we initialize a learnable prompt token $\mathbf{p}_i \in \mathbb{R}^d$ for each $i \in \mathcal{V}$, where $d$ denotes the dimension of latent embedding space. The environment prompt tokens $\mathbf{P}$ for $\mathcal{G}$ is then denoted as,

$$\mathbf{P} = \{\mathbf{p}_i\}_{i=1}^N \in \mathbb{R}^{N \times d}. \tag{1}$$

Since $\mathbf{P}$ is specified on the spatial perspective, the node across different temporal snapshots shares the same environment prompt. In the implementation, $\mathbf{P}$ is expanded as a new tensor of $\mathbb{R}^{T \times N \times d}$ by cloning $T$ times.

**Prompt-answer mechanism for environment squeezing.** The next challenge to address is how to effectively utilize the well-designed prompts to guide the model in extracting underlying environment representations $\mathbf{Z}_E \in \mathbb{R}^{T \times N \times d}$. The premise of achieving this goal lies in profoundly understanding the relationships between the variables in dynamic graph. The ability of Structural Causal Model (SCM) to describe the relationship between variables offers a valuable framework for our analysis [26, 27]. The SCM of dynamic graph includes the temporal environment information $\mathbf{E}$, spatial context $\mathbf{C}$, historical observation $\mathbf{X}$, and future evolving signals $\mathbf{Y}$. Actually, $\mathbf{X}$ denotes historical observations, which is the combination of $\mathbf{X_X}$ and $\mathbf{Y_X}$ defined in Sec. 2. This causal model can be formalized as,

$$\mathbb{P}(\mathbf{X}, \mathbf{Y}|\mathbf{E}, \mathbf{C}) = \mathbb{P}(\mathbf{Y}|\mathbf{X}, \mathbf{E}, \mathbf{C})\mathbb{P}(\mathbf{X}|\mathbf{E}, \mathbf{C}). \tag{2}$$

Therefore, we aim to squeeze environment variables $\mathbf{E}$ from the observable feature $\mathbf{X_X}$ and the observed labels with evolution patterns $\mathbf{Y_X}$, which is similar to solving cloze problems. To accomplish this, we design an interactive squeezing mechanism $g_\theta(\cdot)$ to guide the model to squeeze out the underlying environment variables through the interaction of learnable prompts $\mathbf{P}$ and the observable features $\mathbf{X_X}$. Specifically, we first perform a spatio-temporal network backbone $h_\phi(\cdot)$ to get nodes embedding $\mathbf{Z} = \{\mathbf{z}_1^t, \cdots, \mathbf{z}_N^t\}_{t=1}^T \in \mathbb{R}^{T \times N \times d}$ by taking observable features $\mathbf{X_X}$ as inputs. Then, $g_\theta(\cdot)$ decodes the learnable prompts $\mathbf{P}$ and encoded embedding $\mathbf{Z}$ to obtain prompt answers $\mathbf{Z}_E$. In implementation, $g_\theta(\cdot)$ consists of three families of learnable parameters, i.e., $W^Q, W^K, W^V \in \mathbb{R}^{T \times d \times d}$. Three hidden state matrices are calculated by,

$$\mathbf{P}^Q = \mathbf{P}W^Q, \ \mathbf{Z}^K = \mathbf{Z}W^K, \ \mathbf{Z}^V = \mathbf{Z}W^V. \tag{3}$$

The prompt answer $\mathbf{Z}_E$ is obtained by,

$$\mathbf{Z}_E = \text{softmax}(\frac{\mathbf{P}^Q(\mathbf{Z}^K)^T}{\sqrt{d}})\mathbf{Z}^V + \boldsymbol{\epsilon}, \tag{4}$$

where $\epsilon \sim \mathcal{N}(\mathbf{0}, \boldsymbol{I})$. Random noise is added to further enhance the robustness of inferred environment representation. Then, the ground-truth $\mathbf{Y_X}$ serves as the learning goal during this squeezing process. We design a tractable objective to squeeze unseen environment representation $\mathbf{Z}_E$,

$$\min_{\phi,\theta,\mathbf{P}} \mathcal{L}_P = -\mathbb{E}[\log \mathbb{P}(\mathbf{Y_X}|\mathbf{X_X}, \mathbf{E})] = \beta\mathbb{E}[\mathrm{KL}(\mathbb{P}_\theta(\mathbf{Z}_E)||\mathbb{P}_\phi(\mathbf{Z}))] - \mathbb{E}[\log \mathbb{P}_{\phi,\theta}(\mathbf{Y_X}|\mathbf{Z}_E)], \quad (5)$$

where $\beta \in [0, 1]$ is a preset hyperparameter, and its sensitivity analysis is provided in Appendix G.1.

- The first term captures the similarity between environment states $\mathbf{Z}_E$ and node embedding $\mathbf{Z}$.
- The second term predicts the evolution rule $\mathbf{Y_X}$ only using inferred unseen environments $\mathbf{Z}_E$.

The objective of Eq. 5 implies that our prompt answers not only reflect the evolution of dynamic graphs but also significantly differ from current available features.

Our design infers unseen environment factors into a continuous space, which remarkably distinctive from previous methods with predefined and discrete environment scale [53, 58]. Our extracted $\mathbf{Z}_E$ can eliminate the bias stemming from inadequate prior knowledge of environment information.

**Provable Squeezed Environment Answer.** SPL enables the awareness of environment factors via employing environment prompt framework. However, there is still indistinctness about the feasibility of our design. From the perspective of information theory [2, 5], the learning objective of SPL can be restated as,

$$\max_{\phi,\theta,\mathbf{P}} I(\mathbf{Z}_E; \mathbf{Y_X}) - \beta I(\mathbf{Z}_E; \mathbf{X_X}). \quad (6)$$

We will replace Eq. 5 with above Eq. 6 for later proof.

**Theorem 3.1.** *If there exists a causal relationship between unseen environment pattern $\mathbf{Z}_E^*$ and the label $\mathbf{Y}_X$, $\mathbf{Z}_E^*$ is the optimal result of SPL objective.*

Theorem 3.1 demonstrates the feasibility of SPL and we can always extract additional environment factor if it exist. Detailed proof is provided in Appendix C.

## 3.2 Spatial-temporal Learning with Dynamic Subgraph

With the well-learned environment answers, how to exploit such representations to achieve spatio-temporal prediction becomes a natural problem. Existing environment-centered approaches tend to directly treat the perceived environment embedding $\mathbf{Z}_E$ as the additional feature. However, from the perspective of data generation, the influence of environments on data distribution shifts is usually reflected in the changes of node-wise correlations. Current methods fail to capture the causal effects of environment variables in dynamic graphs. In this

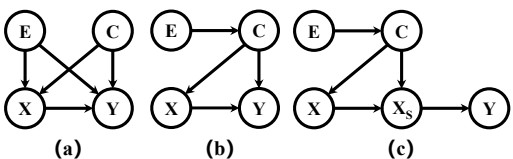

Figure 2: SCMs of dynamic graph. (a) Traditional generation understanding of dynamic graph; (b) Indirect effect of environment factors; (c) Dynamic subgraph as mediating variable.

subsection, we introduce a spatio-temporal invariant learning approach using node-centered dynamic subgraph as the mediating variable.

**A winding causal path in dynamic graph.** We revisit the role of $\mathbf{E}$ in the spatio-temporal causal model, and propose a novel SCM with dynamic subgraph $\mathbf{X_S}$ as mediating variable as shown in Fig. 2(c). It can be formalized by,

$$\mathbb{P}(\mathbf{Y}, \mathbf{X}|\mathbf{E}, \mathbf{C}) = \mathbb{P}(\mathbf{Y}|\mathbf{X_S})\mathbb{P}(\mathbf{X_S}|\mathbf{X}, \mathbf{C})\mathbb{P}(\mathbf{X}|\mathbf{E}, \mathbf{C}). \quad (7)$$

In contrast to the symmetric correlations between nodes in static graphs, the dependencies between nodes in dynamic graphs are often directional and asymmetric. For example, there is a flow direction between nodes in traffic network and certain node pairs may have different influence or importance to each other in social network. As a result, the shift of environments within dynamic graphs always leads to the changes of these asymmetric correlations between nodes. A typical example is that the change of weather conditions always leads to a shift in the direction of traffic flow instead of bring any new paths. Different from partition-based subgraph learning strategies investigated in static

graphs, we design node-centered dynamic subgraph extractor tailored for dynamic graphs, where each node has its unique subgraph based on its environment states. Such strategy not only captures the impact of environments on internode dependencies, but also facilitates to mine the invariant pattern of spatio-temporal evolution more interpretably.

**Dynamic subgraph extraction for environment enhancement.** We argue that the asymmetry of environment factor between nodes often leads to clustering effects, such as the asymmetrical importance between the prominent individuals and their followers brings stable connectivity. We leverage the relative entropy (Kullback-Leibler Divergence) to measure such asymmetry. Relative entropy is a metric employed to quantify the disparity between two probability distributions, which exhibits a notable feature of asymmetry, i.e., $\mathrm{KL}(\mathbb{P}||\mathbb{Q}) \neq \mathrm{KL}(\mathbb{Q}||\mathbb{P})$. $\mathrm{KL}(\mathbb{P}||\mathbb{Q})$ quantifies the degree of match when using $\mathbb{P}$ as the reference distribution and approximating it with $\mathbb{Q}$, while $\mathrm{KL}(\mathbb{Q}||\mathbb{P})$ quantifies the degree of match when using $\mathbb{Q}$ as the reference distribution and approximating it with $\mathbb{P}$. Such property of relative entropy offers significant advantages in quantifying asymmetric environment distributions. This is because the asymmetric dependencies of environment distributions also considers the influence of one environment factor on another as the reference basis. However, computing the KL divergence between every pair of nodes $\mathrm{KL}(\mathbf{Z}_{E(i,:)}^t||\mathbf{Z}_{E(j,:)}^t)$ is computationally intensive. To this end, we propose linear complexity quantification method using the mean distribution of node-level environment distributions as a proxy. $\mathrm{S}_{(i,j)}^t$ represents the dependence from $i$ on $j$ at time $t$,

$$\mathrm{S}_{(i,j)}^t = \mathrm{KL}(\bar{\mathbf{Z}}_E^t||\mathbf{Z}_{E(i,:)}^t) \times \mathrm{KL}(\mathbf{Z}_{E(j,:)}^t||\bar{\mathbf{Z}}_E^t), \qquad (8)$$

where $\bar{\mathbf{Z}}_E^t = \mathrm{MEAN}(\mathbf{Z}_E^t)$ denotes the mean distribution of node-level environment embedding. This method utilizes $\bar{\mathbf{Z}}_E^t$ as an intermediary to measure the environment dependency from node $i$ to $j$. The larger of $\mathrm{S}_{(i,j)}^t$ indicates a greater gap of environment difference from node $i$ to $j$, which reflects the strong dependence. The asymmetric correlation matrix $\mathrm{S}^t \in \mathbb{R}^{N \times N}$ at time $t$ can be calculated as,

$$\mathrm{S}^t = (\boldsymbol{M}^t)^{\mathrm{T}} \cdot \boldsymbol{N}^t. \qquad (9)$$

$\boldsymbol{M}^t \in \mathbb{R}^{1 \times N}$ and $\boldsymbol{N}^t \in \mathbb{R}^{1 \times N}$ are respectively calculated from the two terms in Eq. 8,

$$\boldsymbol{M}^t = [\mathrm{KL}(\bar{\mathbf{Z}}_E^t||\mathbf{Z}_{E(1,:)}^t), \mathrm{KL}(\bar{\mathbf{Z}}_E^t||\mathbf{Z}_{E(2,:)}^t),..., \mathrm{KL}(\bar{\mathbf{Z}}_E^t||\mathbf{Z}_{E(N,:)}^t)], \qquad (10)$$

$$\boldsymbol{N}^t = [\mathrm{KL}(\mathbf{Z}_{E(1,:)}^t||\bar{\mathbf{Z}}_E^t), \mathrm{KL}(\mathbf{Z}_{E(2,:)}^t||\bar{\mathbf{Z}}_E^t),..., \mathrm{KL}(\mathbf{Z}_{E(N,:)}^t||\bar{\mathbf{Z}}_E^t)]. \qquad (11)$$

$\mathrm{S}_{(:,i)}^t$ denotes potential nodes centered on node $i$ and $\mathrm{KL}(\cdot||\cdot)$ indicates the Kullback-Leibler divergence. We can extract node-centered $L$-hop subgraph $\widehat{\mathcal{A}^t} \in \mathbb{R}^{N \times N}$ based on this environment-enhanced correlation matrix $\mathrm{S}^t$,

$$\widehat{\mathcal{A}^t} = \mathrm{S}^t \odot \mathbb{I}((\mathcal{A}^t)^L), \qquad (12)$$

where $\odot$ denotes element-wise multiplication of matrices and $L$ is a hyperparameter. $L$ is set to 5 and its sensitivity is discussed in Appendix G.1. $\mathbb{I}(\cdot)$ is an indicator function that assigns a value of 1 to elements in the matrix that are greater than 0, and assigns a value of 0 to the rest. We can get a series of dynamic subgraphs that evolves over time $\widehat{\mathcal{A}} \in \mathbb{R}^{T \times N \times N}$. Meanwhile, we combine the features of each node by concatenating environment answer $\mathbf{Z}_E^t$ to obtain enhanced $\widehat{\mathcal{X}^t} \in \mathbb{R}^{N \times (D+d)}$,

$$\widehat{\mathcal{X}^t} = \mathrm{CONCAT}([\mathcal{X}^t, \mathbf{Z}_E^t]). \qquad (13)$$

The historical data enhanced by dynamic subgraph with environment representation is denoted as,

$$\mathbf{X_S} = \{\widehat{\mathcal{G}^t}\}_{t=1}^T = \{\mathcal{V}^t, \widehat{\mathcal{X}^t}, \widehat{\mathcal{A}^t}\}_{t=1}^T. \qquad (14)$$

**Dynamic graph prediction with generalizability.** Finally, $f_\psi$ encodes the dynamic subgraphs $\mathbf{X_S}$ via a spatio-temporal network backbone to predict future dynamic evolution. We obtain the learning objective of EpoD,

$$\min_{\phi,\theta,\mathbf{P},\psi} \mathcal{L} = -\mathbb{E}[\log \mathbb{P}_\psi(\mathbf{Y}|\mathbf{X_S}) - \log \mathbb{P}_{\phi,\theta}(\mathbf{Y_X}|\mathbf{X_X}, \mathbf{P})] - \beta\mathbb{E}[\mathrm{KL}(\mathbb{P}_\theta(\mathbf{Z}_E)||\mathbb{P}_\phi(\mathbf{Z}|\mathbf{X_X}))]. \qquad (15)$$

We provide the training process of EpoD in Alg. 1. It is worth noting that our EpoD systematically resolves the limitations of environment inference and environment utilization faced by spatio-temporal invariant learning methods. The two aspects of the design are not independent, but rather tightly coupled. Moreover, our EpoD is pluggable that can be flexibly integrated with various backbones. In

Table 1: The performance of traffic prediction tasks $(12 \rightarrow 24)$ on four real-world datasets. The best results are shown in **bold** and the second best results are underlined.

| Model | PEMS08 | | PEMS04 | | SD(2019-2020) | | GBA(2019-2020) | |
|---|---|---|---|---|---|---|---|---|
| | MAE | RMSE | MAE | RMSE | MAE | RMSE | MAE | RMSE |
| GWNET | 19.04±0.9 | 29.02±1.1 | 23.12±0.8 | 38.75±1.3 | 30.22±2.1 | 43.65±2.9 | 31.27±2.6 | 45.29±2.3 |
| AGCRN | 17.30±0.2 | 27.44±0.6 | 21.19±0.3 | 34.65±0.2 | 26.19±1.2 | 40.51±1.3 | 28.74±1.6 | 43.75±2.0 |
| Z-GCNETs | 19.24±0.3 | 28.40±0.2 | 22.55±0.5 | 36.27±0.7 | 28.21±1.7 | 41.32±1.8 | 29.87±1.2 | 43.11±2.2 |
| DSTAGNN | 17.56±0.3 | 26.29±0.2 | 21.22±0.7 | 36.65±0.2 | 26.34±1.4 | 41.31±1.6 | 30.11±2.0 | 42.99±2.7 |
| STGNCDE | 18.41±0.6 | 27.38±0.3 | 22.04±0.6 | 35.39±0.4 | 27.34±0.9 | 40.73±1.3 | 29.21±1.5 | 43.03±2.4 |
| CaST | 17.28±0.3 | 26.56±0.4 | **20.79±0.4** | 34.95±0.3 | 25.38±1.1 | 39.92±1.6 | 28.67±1.8 | 42.23±1.9 |
| EopD(ours) | **16.92±0.2** | **25.66±0.6** | 21.12±0.4 | **34.02±0.3** | **23.58±1.2** | **38.25±1.4** | **27.26±1.5** | **40.14±1.8** |

the experiments of traffic flow and social relationship prediction, we select Adaptive Graph Convolutional Recurrent Network (AGCRN) [3] and Disentangled Dynamic Graph Attention Networks (DDGAN) [61] as our STGNN backbone respectively.

Our approach does not require predictions of future unseen environments. Actually, we argue that the underlying unseen environment factors within historical observations harbors valuable information to guide evolution. Therefore, our focus lies in perceiving historical environment factors and exploiting them appropriately to capture evolution-invariant pattern for prediction. The subsequent experimental discussion can further validate such intuition.

### 3.3 Causal Interpretation of Dynamic Subgraphs

In this subsection, we provide a deeper understanding of EpoD in the causal theory perspective [28].

**The mediating effect in the dynamic graph.** The efforts on how temporal environment factors and spatial contexts influence the evolution of dynamics graph has been extensively made. However, even though some studies claim disentanglement of spatial-temporal dependencies, it is acknowledged that true separation may not be fully achieved. Most of them inherently concentrated on exploring the interplay between spatial and temporal dynamics. In fact, some pioneering researches have revealed the temporal evolution mostly stem from the changes over spatial dependencies. To this end, we summarize such indirect influence as mediating effect within dynamic graph, as shown in Fig. 2(b). But according to *the complete mediation effect theorem*, this SCM eliminates the direct effect of temporal variable $\mathbf{E}$ on future spatio-temporal evolution $\mathbf{Y}$. This requires us to contemplate whether $\mathbf{C}$ can serve as a mediation variable. Given the time-varying property of dynamic graph, the only spatial context cannot sufficiently interpret the graphs evolution. Therefore, a mediating variable simultaneously encapsulating spatial dependencies and temporal dynamics is required.

**Dynamic subgraphs as mediation variable.** The dynamic spatial variations induced by environment factors are essentially rooted in the changes of local dependencies. Thus, a novel SCM is introduced, which employ dynamic subgraph $\mathbf{X_S}$ as the mediation variable as illustrated in Fig. 2(c). Dynamic subgraphs exhibit both temporal and spatial characteristics, also serve as the mediation variable from $\mathbf{X}$ to $\mathbf{Y}$. This design offers us a chance to address distribution shift issue along the practices of causal adjustment [28]. We can observe a back-door path between causal path $\mathbf{X}$ and $\mathbf{Y}$, i.e., $\mathbf{X} \leftarrow \mathbf{C} \rightarrow \mathbf{X_S} \rightarrow \mathbf{Y}$. The backdoor adjustment pattern leveraging *do-calculus* on dynamic subgraph $\mathbf{X_S}$ is,

$$P(\mathbf{Y} = y | do(\mathbf{X} = x)) = \sum_{x_S} P(\mathbf{Y} = y | \mathbf{X} = x, \mathbf{X_S} = x_S) P(\mathbf{X_S} = x_S). \tag{16}$$

In essence, our EpoD can be viewed as employing backdoor adjustments to estimate $P(\mathbf{Y}|do(\mathbf{X}))$ by discovering dynamic subgraphs, where the prompted environment representations support the subgraph discovery process. More discussion is provided in Appendix D.

## 4 Experiments

### 4.1 Experiment Setup

We introduce datasets, baselines and experiment settings, and details are leaved in Appendix E.

**Datasets.** We employ seven cross-domain real-world dynamic graph datasets to evaluate our EpoD. PEMS08 and PEMS04 [34] are classic medium-scale traffic network datasets from California with 5-minute intervals; SD and GBA [24] are newly proposed large-scale traffic network datasets. COL-LAB [40] is an academic collaboration dataset comprising papers published in 16 years; Yelp [33] is a business review dataset; ACT [18] shows students' actions on a MOOC platform over 30 days.

**Baselines.** We compare EpoD with two families of baselines, i.e., six traffic flow prediction models and six social link forecasting methods. Traffic flow prediction models: GWNET [52], AGCRN [3], Z-GCNETs[8], DSTAGNN [19], STGNCDE [9], CaST [53]. Social link forecasting models: DySAT[33], IRM[1], VREx[17], GroupDRO[32], DIDA[58], EAGLE [58].

**Experiment settings.** In the experiments of traffic flow prediction, our task is to predict the next 24 steps based on historical 12 steps observations ($12 \rightarrow 24$). Moreover, we choose traffic data from the SD and GBA datasets spanning from 2019 to 2020 in order to add the distribution shift scenarios arising from COVID-19, where the training set is composed of data from 2019, while the data from 2020 is divided into a validation set and a test set. The task of social relationship analysis is to exploit past graphs to make link prediction in the next time step. In the training stage, we selectively mask a shifted attribute link from COLLAB, Yelp and ACT to simulate the distribution shift scenario encountered in the real world [61].

## 4.2 Performance Analysis on Real-world Datasets

**Traffic flow prediction.** We evaluate our EpoD with baselines based on Mean Absolute Error (MAE) and Root Mean Square Error (RMSE), where lower values of them represent better performance. Tab. 1 shows the results of EpoD on traffic flow data. We have two observations: 1) our EpoD outperforms all baselines on three datasets. The powerful long-sequence prediction ability of EpoD demonstrates that our design is robust to environment perturbations and excels in capturing the evolution patterns of dynamic graph. We also note that CaST [53] obtains suboptimal results on most datasets and even optimal results on PEMS04. This indicates that it is effective to tackle the temporal distribu-

Table 2: AUC score (%) of future link prediction task on real-world social relationship datasets. The best results are shown in **bold** and the second best results are underlined.

| Model | Collab | Yelp | ACT |
|---|---|---|---|
| DySAT | 76.59±0.20 | 66.09±1.42 | 66.55±1.21 |
| IRM | 75.42±0.87 | 56.02±16.08 | 69.19±1.35 |
| VREx | 76.24±0.77 | 66.41±1.87 | 70.15±1.09 |
| GroupDRO | 76.33±0.29 | 66.97±0.61 | 74.35±1.62 |
| DIDA | 81.87±0.40 | 75.92±0.90 | 78.64±0.97 |
| EAGLE | **84.41±0.87** | 77.26±0.74 | 82.70±0.72 |
| EopD(ours) | 83.21±0.35 | **80.85±0.81** | **83.85±0.52** |

tion shift issue by studying environment factors under the guidance of causal theory. 2) EpoD exhibits a more pronounced capability for prediction improvement on two large-scale traffic datasets. It indicates that EpoD is better suited for addressing the distribution shift issue caused by extremely intricate environment perturbations, which is the main challenge posed by large-scale traffic data. We also discuss the interpretability of dynamic subgraphs in Appendix G.3.

**Social link forecasting.** Tab. 2 presents the performance of EpoD on social link prediction tasks. Our model outperforms all baselines on two datasets under distribution shifts. We also observe that EAGLE [58] achieves one best performance and other sub-optimal results, which is comparable to our method. It proves that the approaches of perceiving environments can tackle the distribution shifts issue in dynamic graph. Besides, our extracted continuous environment representations are more expressive than the environment factors with pre-defined scales.

## 4.3 Toy Dataset

We manually design a toy dataset *EnvST* with temporal distribution shift to explore the generalizability of EpoD. The feature of each node in *EnvST* encompasses three components, i.e., $[x_A, x_B, x_C]$, where $x_A$ and $x_B$ represent evolution-causal features but $x_B$ is masked after the data is generated, $x_C$ indicates available evolution-spurious feature. To simulate the temporal distribution shift in the dynamic graph, the training and test dataset of $x_A$, $x_B$ and $x_C$ are sampled from distributions with significant differences separately. The label of each node on the *EnvST* at $t$ step is activated by updated feature $y_i^t \sim \text{Bern}(\sigma(z_i^t))$, where $\sigma(\cdot)$ is the sigmoid func-

tion. We conduct experiments from the following two aspects: 1) we investigate whether EpoD has the capability to perceive masked environment feature $x_B$, 2) we study whether EpoD can identify and remove the spurious correlation $x_C$. More analysis can be found in Appendix E.4.

**Powerful perception for unseen environment.**
Fig. 3(a) and 3(b) show the distribution difference between masked feature $x_B$ and prompted environment feature $\mathbf{Z}_E$, where experiments are conducted on *EnvST* under the scenario of distribution shift. Fig. 3(a) shows our prompted environment feature $\mathbf{Z}_E$ can cover more than half of the shifted features in the future steps. As shown in Fig. 3(b), we observe that our prompted environment variables can effectively cover slight early signal and utilize it to tackle OOD issue.

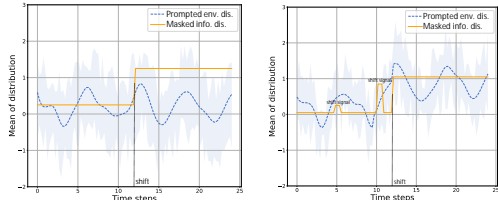

(a) Sharp temporal distribution shift. (b) Temporal distribution shift with slight signals.

Figure 3: Analysis on the toy dataset.

**Robust spurious information identification ability.** We then explore whether our EpoD can filter out the disturbance of $x_C$. Specifically, we have the following experiment design. Consider $x_C$ is sampled from $\mathcal{N}(\mu, I)$, we set $\mu \in [0, 10]$ and record the performance of EpoD under the influence of different spurious information as shown in Fig. 7. We can observe that the fluctuations in prediction performance consistently fall within the acceptable error bounds. Therefore, we can conclude that EpoD have the ability to identify spurious information $x_C$.

## 4.4 Ablation Study

We conduct ablation studies from the following two aspects.

**Temporal shared learnable prompts.** In spatio-temporal graphs, we can observe different nodes always reveal heterogeneous evolution patterns. Thus, we design temporal shared node-wise learnable prompts. In essence, this design is driven by both resource consumption and real-world scenarios. There are still two potentially effective design approaches: a single learnable prompt shared globally (SingleP) and node-private learnable prompts (PrivateP). The former only initializes a globally shared prompt for the dynamic graph $\mathbf{P} \in \mathbb{R}^d$; the latter assigns learnable prompts to each node of each snapshot $\mathbf{P} \in \mathbb{R}^{T \times N \times d}$. To this end, we compare three design methods in terms of accuracy and efficiency. Fig. 4 shows the results of ablation, where the bars represent the time consuming and the lines depict accuracy. We have the following two observations.

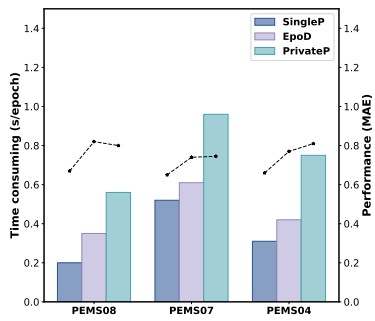

Figure 4: A comparison of learnable prompts design approaches.

1) From the aspects of accuracy, the design of EpoD and PrivateP have similar performance, and SingleP is significantly inferior to them. 2) From the aspects of efficiency, it is understandable that SingleP has the highest training efficiency and PrivateP is the least efficient. The efficiency of EpoD falls between them, yet it remains competitive with SingleP. Therefore, we can conclude that our design stands out as the optimal choice considering both efficiency and accuracy.

**The necessity of dynamic subgraph design.** We aim to investigate the importance of dynamic subgraphs and explore which subgraph extraction strategy is more suitable for dynamic graph learning. Specifically, we first construct two variants of EpoD, i.e., EpoD-NoSub and EpoD-PartitionSub. EpoD-NoSub is a variant of EpoD that does not utilize dynamic subgraphs for spatiotemporal prediction, which just uses the perceived environment only by incorporating features.

Table 3: Ablation results on dynamic subgraph in EpoD. MAE performance on PEMS08 and PEMS04, AUC(%) score on Yelp.

| Model | PEMS08 | PEMS04 | Yelp |
|---|---|---|---|
| EpoD-NoSub | 17.45 | 21.93 | 76.34 |
| EpoD-PartitionSub | 18.09 | 22.04 | 77.35 |
| EopD(ours) | **16.92** | **21.12** | **80.85** |

EpoD-PartitionSub means an EpoD variant with partition-based subgraph extraction strategy like static graph learning. We perform ablation experiments on PEMS08, PEMS04 and Yelp, as shown in Tab. 3. First, we observe that the performance difference between EpoD and EpoD-NoSub is substantial, with a maximum gap exceeding 2. It means the utilization of dynamic subgraphs not only enhances interpretability but also is crucial for improving generalization performance. Second, we EpoD had a more pronounced effect than EpoD-PartitionSub. In addition, more experiments show that EpoD is more stable than EpoD-PartitionSub.

### 4.5 Efficiency Analysis

We analyze the efficiency of EpoD theoretically and practically. We utilize $|V|$ and $|E|$ to denote the number of nodes and edges in the graph, $d$ to represent the dimension of implicit representation, and $T$ to represent the time step of historical observations. The time consumption mainly comprises three components: the spatio-temporal graph aggregation process, the prompt answer process, and the dynamic subgraphs sampling process. The time complexity of the spatio-temporal aggregation is $O(T \cdot (|E| \cdot d + |V| \cdot d^2))$. The prompt answer

Table 4: Efficiency analysis of EpoD (s/epoch).

| Dataset | DIDA | EAGLE | EpoD |
|---|---|---|---|
| COLLAB | 11.21 | 12.05 | 11.84 |
| Yelp | 6.89 | 7.38 | 7.34 |
| ACT | 9.27 | 9.76 | 9.59 |

process primarily involves a cross-attention operation, with a time complexity of $O(T \cdot |V| \cdot d)$. The dynamic subgraphs sampling module implements node-centered sampling, with a time complexity of $O(T \cdot |V|)$. Therefore, the time complexity of EpoD is $O(T \cdot d \cdot |E| + T \cdot (1 + d + d^2) \cdot |V|)$. In conclusion , EpoD exhibits linear time complexity concerning the number of nodes and edges, which is competitive with existing dynamic GNNs such as DIDA, EAGLE, and CaST.

We also conduct efficiency comparisons of EpoD, DIDA, and EAGLE in COLLAB, Yelp, and ACT datasets, measuring the time taken per epoch (s/epoch). All experiments are run on an NVIDIA A100-PCIE-40GB. Empirically, we observe that the operational efficiency of our method is competitive with existing approaches.

## 5 Conclusion and Future Work

In this paper, we propose a novel dynamic graph learning framework EpoD to tackle the temporal distribution shift issue by exploiting prompt learning. Inspired by the powerful ability of prompt learning in perceiving underlying semantic and causal associations, we first introduce a self-prompted environment inference mechanism. This approach aims to extract underlying environment variables that potentially influence temporal distribution shift. Subsequently, we propose a novel causal pathway that leverages dynamic subgraphs as mediating variables to effectively utilize the inferred environment embedding. Experiments on real-world datasets and toy examples show that our EpoD effectively improve the dynamic graph learning under temporal shifts, especially boosting the interpretability via dynamic subgraphs.

**Limitations.** One of the limitation of our work is its strong dependence on graph topology. Specifically, our subgraph discovery strategy essentially is the node filtering based on K-hop neighbors, which can be regarded as subtracting elements from the graph topology. However, the graph topology constructed by distance-based adjacency matrix always lacks adaptability to dynamic changes in the relationships between nodes [53]. In the future, we aim to improve the extraction process of node-centered dynamic subgraphs. We intend to form subgraphs from an empty topology by taking into account both the perceived environment embeddings and the initial distance information.

## Acknowledgements

This paper is partially supported by the National Natural Science Foundation of China (No.12227901, No.62072427, No.62402414), Natural Science Foundation of Jiangsu Province (BK.20240460), the Project of Stable Support for Youth Team in Basic Research Field, CAS (No.YSBR-005), Academic Leaders Cultivation Program, USTC, and the grant from State Key Laboratory of Resources and Environmental Information System. We sincerely thank all reviewers for their insightful and constructive comments in improving this paper.

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

# A  Broader impacts

Dynamic graph learning models are widely used to support social development, such as recommendation systems and smart cities. However, with the increasing complexity of data scale and application scenarios, the distribution shifts between training and test data have become a significant obstacle in the development of dynamic graph learning. In light of this, our work aims to address the issue of data distribution shifts in the model and promote the broader application of graph learning in various fields. Therefore, our work aims to develop a model with the out-of-distribution generalization ability and thereby promote the widespread application of dynamic graph learning in various fields.

We ensure the full ethical compliance of our work, and all the datasets we utilize are publicly available. Our work does not involve human subjects and does not introduce any potential negative social impacts or issues related to privacy and fairness.

# B  Notations

Table 5: Classification accuracies for naive Bayes and flexible Bayes on various data sets.

| NOTATIONS | DESCRIPTIONS |
|---|---|
| $\mathcal{G} = \{\mathcal{G}^t\}_{t=1}^T$ | A DYNAMIC GRAPH ACROSS $T$ STEPS |
| $\mathcal{G}^t = (\mathcal{V}^t, \mathcal{X}^t, \mathcal{A}^t)$ | A $t$-STEP GRAPH WITH THE NODES $\mathcal{V}^t$, FEATURES $\mathcal{X}^t$ AND EDGES $\mathcal{A}^t$ |
| $\mathcal{X}^t \in \mathbb{R}^{N \times D}$ | THE FEATURE MATRIX $\mathcal{X}^t$ OF $t$-STEP GRAPH SNAPSHOT |
| $\mathbf{P} \in \mathbb{R}^{T \times N \times d}$ | LEARNABLE PROMPTS MATRIX OF $\mathbb{R}^{T \times N \times d}$ |
| $\mathbf{X} = (\mathbf{X_X}, \mathbf{Y_X})$ | HISTORICAL OBSERVATIONS IN PREDICTION TASKS |
| $\mathbf{X_S} = \{\mathcal{V}^t, \widehat{\mathcal{X}}^t, \widehat{\mathcal{A}}^t\}_{t=1}^T$ | HISTORICAL OBSERVATIONS REPRESENTED BY DYNAMIC SUBGRAPHS |
| $\mathbf{E}$ | TEMPORAL ENVIRONMENT VARIABLE IN PREDICTION TASKS |
| $\mathbf{Y}$ | THE FUTURE EVOLUTION TREND IN PREDICTION TASKS |
| $\mathbf{X_X}$ | THE OBSERVABLE FEATURE OF HISTORICAL OBSERVATIONS |
| $\mathbf{Y_X}$ | THE OBSERVED LABELS WITH EVOLUTION PATTERNS |
| $\mathbf{Z}_E$ | CONTINUOUS FEATURES OF ENVIRONMENT VARIABLES |
| $\mathbf{Z}$ | NODE EMBEDDING BY ENCODING ORIGINAL FEATURES |
| $\mathbf{C}$ | SPATIAL VARIABLE IN STRUCTURAL CAUSAL MODEL |
| $g_\theta(\cdot)$ | A CROSS-ATTENTION NETWORK |
| $h_\phi(\cdot)$ | A SPATIO-TEMPORAL NETWORK BACKBONE |
| $f_\psi(\cdot)$ | A SPATIO-TEMPORAL NETWORK BACKBONE IN FINAL PREDICTION STAGE |

# C  Detailed Proof of Theorem 3.1

**Lemma 3.1.** (*The Chain Rule for Mutual Information* [2]) For any set of random variables $X$, $Y$, and $Z$, the chain rule is expressed as

$$I(X; Y, Z) = I(X; Y) + I(X; Z|Y), \tag{17}$$

where $I(X; Y, Z)$ denotes the mutual information between $X$, $Y$, and $Z$, $I(X; Y)$ represents the mutual information between $X$ and $Y$, and $I(X; Z|Y)$ represents the conditional mutual information between $X$ and $Z$ given $Y$. The rule signifies that the overall mutual information in the system can be decomposed into two components: the mutual information between $X$ and $Y$, and the conditional mutual information between $X$ and $Z$ given $Y$. It reflects how information is transmitted and shared within complex systems.

**Theorem 3.1.** If there exists a causal relationship between unseen environment pattern $\mathbf{Z}_E^*$ and the label $\mathbf{X_X}$, $\mathbf{Z}_E^*$ is the optimal solution of SPL objective.

*Proof.* Consider *The Chain Rule for Mutual Information*, we derive the following equation:

$$I(\mathbf{Z}_E; \mathbf{Y_X}) = I(\mathbf{Y_X}; \mathbf{X_X}, \mathbf{Z}_E) - I(\mathbf{X_X}; \mathbf{Y_X}|\mathbf{Z}_E) \tag{18}$$

$$I(\mathbf{Z}_E; \mathbf{X_X}) = I(\mathbf{X_X}; \mathbf{Z}_E, \mathbf{Y_X}) - I(\mathbf{X_X}; \mathbf{Y_X}|\mathbf{Z}_E) \tag{19}$$

We then get $I(\mathbf{Z}_E; \mathbf{Y_X}) - \beta I(\mathbf{Z}_E; \mathbf{X_X}) = I(\mathbf{Y_X}; \mathbf{X_X}, \mathbf{Z}_E) - (1-\beta)I(\mathbf{X_X}; \mathbf{Y_X}|\mathbf{Z}_E) - \beta I(\mathbf{X_X}; \mathbf{Z}_E, \mathbf{Y_X})$. Thus, optimizing Eq. 6 is to maximize $I(\mathbf{Y_X}; \mathbf{X_X}, \mathbf{Z}_E)$ and minimize $(1-\beta)I(\mathbf{X_X}; \mathbf{Y_X}|\mathbf{Z}_E) + \beta I(\mathbf{X_X}; \mathbf{Z}_E, \mathbf{Y_X})$. Next, we investigate whether $\mathbf{Z}_E^*$ is the result of optimization.

(i) Maximizing $I(\mathbf{Y_X}; \mathbf{X_X}, \mathbf{Z}_E)$ reflects the combination of $\mathbf{X_X}$ and $\mathbf{Z}_E$ can make optimal prediction. Since $\mathbf{X_X}$ is constant, $\mathbf{Z}_E = \mathbf{Z}_E^*$ ensures that the features used for prediction have the greatest overlap with the ground-truth.

(ii) Since $I(\mathbf{X_X}; \mathbf{Y_X}|\mathbf{Z}_E) \geq 0$ and $I(\mathbf{X_X}; \mathbf{Z}_E, \mathbf{Y_X}) \geq 0$. If $\beta \in [0, 1]$, the lower bound of $(1-\beta)I(\mathbf{X_X}; \mathbf{Y_X}|\mathbf{Z}_E) + \beta I(\mathbf{X_X}; \mathbf{Z}_E, \mathbf{Y_X})$ is 0. Next, we prove our SPL can achieve $I(\mathbf{X_X}; \mathbf{Y_X}|\mathbf{Z}_E^*) = 0$ and $I(\mathbf{X_X}; \mathbf{Z}_E^*, \mathbf{Y_X})) = 0$ in detail. $I(\mathbf{X_X}; \mathbf{Y_X}|\mathbf{Z}_E^*)$ represents the conditional mutual information between $\mathbf{X_X}$ and $\mathbf{Y_X}$ given $\mathbf{Z}_E^*$. Because we manually add random noise $\epsilon$ at Eq. 4, our model can be viewed as a map $\mathbf{Y_X} = f(\mathbf{Z}_E^*) + \epsilon$. Obviously, $\epsilon$ is independent of $\mathbf{Y_X}$. We can get $I(\mathbf{X_X}; \mathbf{Y_X}|\mathbf{Z}_E^*) = 0$. $I(\mathbf{X_X}; \mathbf{Z}_E^*, \mathbf{Y_X})) = 0$ is also proven the independence between $\mathbf{X_X}$ and $\mathbf{Z}_E^*$.

We have proved it.

## D   Causal Interpretation of Dynamic Subgraphs

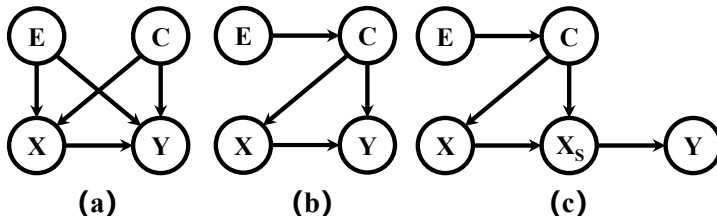

Figure 5: SCMs of dynamic graph. (a) Traditional generation understanding of dynamic graph; (b) Indirect effect of environment factors; (c) Dynamic subgraph as mediating variable.

With the causal theory [28], we can build a coherent progression for the proposal of EpoD.

**The Structural Causal Model in the dynamic graph.** The Structural Causal Model (SCM) in the dynamic graph fosters a deeper understanding of the generation process of spatio-temporal graphs. Many methodologies have achieved impressive performance in addressing the issue of spatial-temporal distribution shifts. The widely utilized SCM is shown in Fig. 5(a).

- $\mathbf{E} \to \mathbf{X} \leftarrow \mathbf{C}$. The historical observation $\mathbf{X}$ consists of two parts: temporal environment variable $\mathbf{E}$ and spatial context $\mathbf{C}$. In many works, these two aspects are often treated as disjoint factors and discussed decouplingly, i.e, $\mathbf{E} \perp \mathbf{C}$.

- $\mathbf{E} \to \mathbf{Y} \leftarrow \mathbf{C}$. Temporal environment variable $\mathbf{E}$ and spatial context $\mathbf{C}$ also determine the future evolution trend $\mathbf{Y}$. Most studies keep this assumption that historical observations $\mathbf{X}$ and future evolution $\mathbf{Y}$ are influenced by the same $\mathbf{E}$ and $\mathbf{C}$. However, this is not the case in the real-world dynamic graphs, giving rise to the distribution shift issue.

- $\mathbf{X} \to \mathbf{Y}$. Historical observations are useful for predicting future evolution. A thorough understanding of historical observations serves as a crucial foundation for exploring the invariant evolution model of dynamic graphs.

**The mediating effect in the dynamic graph.** The efforts on how temporal environment factors and spatial contexts influence the evolution of dynamics graph has been extensively made. However, even though some studies claim disentanglement of spatial-temporal dependencies, it is acknowledged that true separation may not be fully achieved. Most of them inherently concentrated on exploring the interplay between spatial and temporal dynamics. In fact, some pioneering researches have revealed the temporal evolution mostly stem from the changes over spatial dependencies. To this end, we further summarize such indirect influence as mediating effect within dynamic graph, as shown in Fig. 5(b).

- **E → C → X**. The historical observation **X** is seen as the evolution process of spatial context **C**, where **C** covers the temporal environment information **E**.
- **X ← C → Y**. Spatial context **C** directly determine current observations **X** and future evolution trend **Y**.

But according to *the complete mediation effect theorem*, this SCM eliminates the direct effect of temporal variable **E** on future spatio-temporal evolution **Y**. This necessitates us to contemplate whether **C** has the capacity to serve as a mediation variable. Given the time-varying property in dynamic graph, the only spatial context cannot sufficiently interpret the temporal evolution of graphs. Therefore, a mediating variable simultaneously encapsulating spatial dependencies and temporal dynamics is required.

**Dynamic subgraphs as mediating variables.** The dynamic spatial variations induced by environment factors are essentially rooted in the changes of local dependencies. Thus, a novel SCM is introduced, which employ dynamic subgraph $\mathbf{X_S}$ as the mediation variable as illustrated in Fig. 5(c).

- **X ← C → X_S → Y**. Dynamic subgraph $\mathbf{X_S}$ is more abundant than the spatial context **C**. In other words, dynamic subgraph $\mathbf{X_S}$ utilizes substructures to encompass spatial information **C**, capturing temporal environment factors **E** through dynamic evolutive subgraphs.

Dynamic subgraphs exhibit both temporal and spatial characteristics, also serve as the mediation variable from **X** to **Y**. This design offers us a chance to address distribution shift issue along the practices of causal adjustment [28]. We can observe a back-door path between causal path **X** and **Y**, i.e., $\mathbf{X \leftarrow C \rightarrow X_S \rightarrow Y}$. The backdoor adjustment pattern leveraging *do-calculus* on dynamic subgraph $\mathbf{X_S}$ is,

$$P(\mathbf{Y} = y | do(\mathbf{X} = x)) = \sum_{x_S} P(\mathbf{Y} = y | \mathbf{X} = x, \mathbf{X_S} = x_S) P(\mathbf{X_S} = x_S) \qquad (20)$$

In essence, our EpoD can be viewed as employing backdoor adjustments to estimate $P(\mathbf{Y}|do(\mathbf{X}))$ by discovering dynamic subgraphs, where the prompted environment supports the subgraph discovery process.

## E Experiment Details

### E.1 Datasets

Our experimental design included the selection of seven real-world dynamic graph datasets from two distinct domains. The detailed statistics of the datasets are as shown in Tab. 6. We select a shifted link attribute from COLLAB, Yelp and ACT datasets respectively to simulate the distribution shift scenario in the real world. The shifted attribute links become accessible only during the out-of-distribution (OOD) testing stage. This scenario is more practical and challenging in real-world situations, as the model cannot capture any information about the filtered links during training and validation.

**PEMS08** [34] is collected from the Caltrans Performance Measurement System (PeMS), which records the real traffic network flow data from 07/01/2016 to 08/31/2016. It delineates a dynamic graph data of a traffic network with 170 sensors across 17,856 steps. Among the known traffic datasets, it falls into the category of small-scale dataset.

**PEMS04** [34] records the real traffic network flow data from 01/01/2018 to 02/28/2018. It describes a dynamic graph data of a traffic network with 307 sensors across 16,992 steps. It belongs to a medium-scale traffic dataset.

**SD** [24] is a sub-dataset of the large-scale dataset CA proposed by [24]. It comprises traffic flow data recorded by 716 sensors in San Diego county from 01/01/2017 to 12/31/2021.

**GBA** [24] is a larger traffic dataset than SD, which is also a sub-dataset of the large-scale dataset CA. It contains traffic flow data provided by 2,352 sensors in 11 counties situated in the Greater Bay Area from 01/01/2017 to 12/31/2021.

**COLLAB** [40] is an academic collaboration dataset comprising papers published between 1990 and 2006, spanning 16 graph snapshots. In this dataset, nodes represent authors, and edges represent co-authorship relationships. The edges include five attributes based on co-authored publications: "Data Mining", "Database", "Medical Informatics", "Theory" and "Visualization".

**Yelp** [33] is a dataset containing customer reviews on businesses. In this dataset, nodes represent customers and businesses, while edges capture review behaviors. The edges are associated with five attributes based on business categories: "Pizza", "American (New) Food", "Coffee & Tea", "Sushi Bars" and "Fast Food".

**ACT** [18] characterizes student interactions on a MOOC platform over a span of one month, consisting of 30 graph snapshots. In this dataset, nodes represent students or the targets of actions, while edges signify various student actions.

Table 6: Statistics of the real-world dynamic graph datasets.

| DATASET | # NODES | # EDGES | # GRAPH SNAPSHOTS | TEMPORAL INTERVAL |
|---------|---------|---------|-------------------|-------------------|
| PEMS08 | 170 | 276 | 17,856 | 5 MINUTES |
| PEMS04 | 307 | 338 | 16,992 | 5 MINUTES |
| SD | 716 | 17,319 | 525,888 | 5 MINUTES |
| GBA | 2,352 | 61,246 | 525,888 | 5 MINUTES |
| COLLAB | 23,035 | 151,790 | 16 | 1 YEAR |
| YELP | 13,095 | 65,375 | 24 | 1 MONTH |
| ACT | 20,408 | 202,339 | 30 | 1 DAY |

## E.2 Detailed Implementation

---

**Algorithm 1:** The training process of EpoD

---

**Input:** historical dynamic graph data $X = \{\mathcal{G}^t\}_{t=1}^T$
**Initial:** dynamic graph encoders $h_\phi$ and $f_\psi$, cross-attention mechanism decoder $g_\theta$, learnable prompt $\mathbf{P}$, the number of epochs $K$
**for** $i = 1$ **to** $K$ **do**
  **Environment prompt stage:**
  $\mathbf{Z}_E = g_\theta(\mathbf{P}, \mathbf{Z}) + \epsilon$, $\mathbf{Z} = h_\phi(\mathbf{X_X})$ $(\epsilon \sim \mathcal{N}(\mathbf{0}, \mathbf{I}))$
  $\hat{\mathbf{Y}}_{\mathbf{X}} = \text{Linear}(\mathbf{Z}_E)$
  **Environment utilization stage:**
  $\mathbf{S}^t = (\boldsymbol{M}^t)^{\text{T}} \cdot \boldsymbol{N}^t$ as shown in Eq. 9
  $\widehat{\mathcal{A}^t} = \mathbf{S}^t \odot (\mathcal{A}^t)^K$, $\quad \widehat{\mathcal{X}} = \text{CONCAT}([\mathcal{X}, \mathbf{Z}_E])$
  $\mathbf{X_S} = \{\widehat{\mathcal{G}^t}\}_{t=1}^T = \{\mathcal{V}^t, \widehat{\mathcal{X}}^t, \widehat{\mathcal{A}}^t\}_{t=1}^T$
  $\widehat{\mathbf{Y}} = f_\psi(\mathbf{X_S})$
  **Optimizing:**
  $\min_{\phi, \theta, \mathbf{P}, \psi} \mathcal{L} = -\mathbb{E}[\log \mathbb{P}_\psi(\mathbf{Y}|\mathbf{X_S}) + \log \mathbb{P}_{\phi, \theta}(\mathbf{Y_X}|\mathbf{X_X}, \mathbf{P})] + \beta \mathbb{E}[\text{KL}(\mathbb{P}_\theta(\mathbf{Z}_E)||\mathbb{P}_\phi(\mathbf{Z}|\mathbf{X_X}))]$
**end for**
**Return** $h_\phi$, $f_\psi$, $g_\theta$ and $\mathbf{P}$

---

We implement our EpoD with PyTorch 1.11.0 on a server with NVIDIA A100-PCIE-40GB. The detailed training process is shown in Alg. 1. All experiments are repeated with 10 different random seeds of [1,2,3,4,5,6,7,8,9,10]. The reported results include the mean and standard deviation obtained from these 10 runs.

**Traffic flow prediction.** In the experiments of traffic flow prediction, our task is to predict the next 24 steps based on the historical 12 steps observations ($12 \rightarrow 24$). Besides, we choose traffic data from the SD and GBA datasets from 2019 to 2020 in order to add the distribution shift scenarios arising from COVID-19. The selected spatio-temporal graph neural network backbone is Adaptive Graph Convolutional Recurrent Network (AGCRN) [3].

**Social link prediction.** The task of social relationship analysis is to exploit past graphs to make link prediction in the next time step. Following the measures of [61], we introduce perturbations to test data to simulate the scenario of distribution shift in those datasets. Specifically, we select Data Mining and Pizza as the shifted attributes in COLLAB and Yelp. For dataset ACT, we employ K-Means to cluster the action features into five categories and randomly select a certain category (the 5th cluster) of edges as the shifted attribute. Besides, Disentangled Dynamic Graph Attention Networks (DDGAN) [61] is chosen as our spatio-temporal graph neural network backbone.

### E.3  Metrics

We utilize Mean Absolute Error (MAE) and Root Mean Squared Error (RMSE) to assess the performance of our EpoD and baselines. Both metrics quantify the error between model predictions and actual observations in regression tasks. A smaller value for these metrics indicates better model performance. MAE is less sensitive to outliers due to its use of absolute differences, while RMSE is more sensitive to large errors due to its use of squared differences. Given the actual observation $Y_i$ and the corresponding predicted value $\hat{Y}_i$ for $n$ samples, two metrics are calculated as follows:

$$MAE = \frac{1}{n} \sum_{i=1}^{n} |Y_i - \hat{Y}_i| \tag{21}$$

$$RMSE = \sqrt{\frac{1}{n} \sum_{i=1}^{n} (Y_i - \hat{Y}_i)^2}. \tag{22}$$

### E.4  Toy dataset

We manually design a toy dataset *EnvST* with temporal distribution shift to explore the generalizability of EpoD. Tab. 7 describes the statistical information of *EnvST*. Specifically, *EnvST* illustrates the evolution sequence of a graph with 100 nodes across 1000 steps, where the topology of each snapshot does not change over time. The feature of each node in *EnvST* encompasses three components, i.e., $[x_A, x_B, x_C]$, where $x_A$ and $x_B$ represent evolution-causal features but $x_B$ is masked after the data is generated, $x_C$ indicates available evolution-spurious feature. To simulate the temporal distribution shift in the dynamic graph, the training and test dataset of $x_A$, $x_B$ and $x_C$ are sampled from distributions with significant differences separately. The label of each node on the *EnvST* at $t$ step is activated by updated feature $y_i^t \sim \text{Bern}(\sigma(z_i^t))$, where $\sigma(\cdot)$ is the sigmoid function.

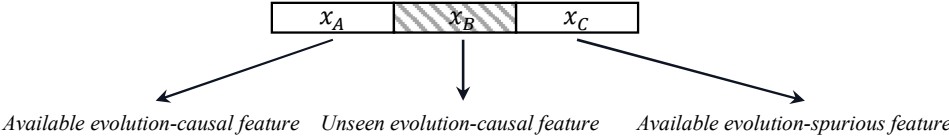

Figure 6: Feature description of the toy dataset.

Table 7: Statistics of the toy dataset.

| NOTATION | DESCRIPTION | DIMENSION |
|---|---|---|
| $N$ | THE NUMBER OF NODES IN THE GRAPH | 100 |
| $T$ | THE NUMBER OF EVOLUTION STEPS OF DYNAMIC GRAPHS | 1000 |
| $x_A$ | AVAILABLE EVOLUTION-CAUSAL FEATURE | 3 |
| $x_B$ | UNSEEN EVOLUTION-CAUSAL FEATURE | 3 |
| $x_C$ | AVAILABLE EVOLUTION-SPURIOUS FEATURE | 3 |
| $y$ | EVOLUTION LABEL | 1 |

As shown in Fig. 6, the feature of each node in *EnvST* encompasses three components, i.e., $[x_A, x_B, x_C]$. $x_A$ and $x_B$ represent evolution-causal features but $x_B$ is masked after the data is generated,

$x_C$ indicates available evolution-spurious feature. To simulate the temporal distribution shift in the dynamic graph, the training and test dataset of $x_A$, $x_B$ and $x_C$ are sampled from distributions with significant differences separately. The label of each node on the *EnvST* at $t$ step is activated by updated feature $y_i^t \sim \text{Bern}(\sigma(z_i^t))$, where $\sigma(\cdot)$ is the sigmoid function. The feature is updated by aggregating the neighbor information of the last $k$ time steps,

$$z_i^t = \sum_{l=t,...,t-k} \sum_{j \in \mathcal{N}_i} \text{Aggregate}(x_A^l(j), x_B^l(j)). \tag{23}$$

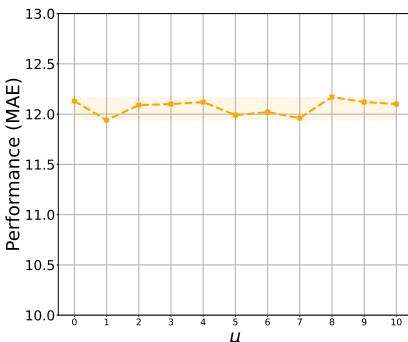

Figure 7: Performance is influenced by spurious information.

We conduct experiments from the following two aspects: 1) we investigate whether EpoD has the capability to perceive masked environment feature $x_B$, 2) we study whether EpoD can identify and remove the spurious correlation $x_C$.

**Powerful perception for unseen environment.** Fig. 3(a) and 3(b) show the distribution difference between masked feature $x_B$ and prompted environment feature $\mathbf{Z}_E$, where experiments are conducted on *EnvST* under the scenario of distribution shift. There is no doubt that $\mathbf{Z}_E$ can match the masked $x_B$ in the steady historical observation sequences, before *shift point* in these figures. More importantly, we focus on the perception ability of prompt learning after the distribution shift occurs. First, we categorize the scenarios of temporal distribution shift into two types: sharp shifts and shifts with signals. The former indicates that any distribution shift signal can not be obtained from historical sequences. Fig. 3(a) shows our prompted environment feature $\mathbf{Z}_E$ can still cover more than half of the shifted features in the future steps. The latter illustrates the signals of distribution shift has begun to emerge slightly in historical observations, which is a commonplace scenario in the real world. For example, those individuals who were used to planning purchased warm clothes in advance before the temperatures plummeted. As shown in Fig. 3(b), we observe that our prompted environment variables can effectively cover slight early signal and utilize it to tackle OOD issue.

**Robust spurious information identification ability.** We then explore whether our EpoD can filter out the disturbance of $x_C$. Specifically, we have the following experiment design. Consider $x_C$ is sampled from $\mathcal{N}(\mu, I)$, we set $\mu \in [0, 10]$ and record the performance of EpoD under the influence of different spurious information as shown in Fig. 7. We can observe that the fluctuations in prediction performance consistently fall within the acceptable error bounds. Therefore, we can conclude that EpoD have the ability to identify spurious information $x_C$.

# F  More Realted Works

**Dynamic Graph Learning.** Graph Neural Networks (GNNs) [10, 37, 54, 44, 36] and Sequence Neural Networks (SNNs) [56, 15] have been extensively studied and have achieved great success in real-world tasks. The GNN models we often use are GCN, GIN, PNA, etc; SNNs include LSTM, RNN, TCN, etc. Therefore, spatio-temporal graph learning models have been widely studied in recent years [50, 46, 45, 41, 43, 42]. Based on varying interpretations of the correlation between temporal and spatial information, the current works are undertaken along two research lines. One claim is to study temporal and spatial information in a decoupled manner, which is potentially present in most current works [3, 19, 61]. Due to the intricate temporal and spatial relationships in reality, they

often fail to offer sufficient interpretability and generalizability. The other one argues that spatial context is influenced by the temporal information [65, 53]. These methods depict spatiotemporal scenes that align more closely with the complexities of the real world. Moreover, this also presents a feasible approach to tackle the distribution shift issue arising from the changes of temporal environment. Our work falls into advancing the latter research line by exploiting causal structure model.

**Subgraph Learning.** Subgraph learning, with its robust causal interpretability, has achieved remarkable success in static graph applications, such as molecular property prediction and social network analysis [25, 57, 38, 21]. In essence, it captures the inductive bias inherent in graph data that local dependencies are invariant patterns predicting ground-truth. We argue that dynamic subgraphs may exhibit a similar inductive bias spatio-temporal evolution. But dynamic subgraphs remain an unexplored area with no existing studies. To this end, this paper introduces a dynamic subgraph learning mechanism to address the issue of temporal distribution shift resulting from the changes of environment factors.

# G    Additional Results

## G.1    Hyperparameter Sensitivity Analysis

We analyze the sensitivity of the hyperparameter $\beta$ in Eq. 5, which functions as the trade-off for the loss in Eq. 5. The value range of $\beta$ is $[0, 1]$. We perform experiments on four real-world datasets, i.e, PEMS08, PEMS04, SD, Yelp, and present the results in Fig 8. The results show that the sensitivity of the prediction results to $\beta$ is not very drastic. But the performances of EpoD on four datasets are the best when $\beta \in [0.2, 0.5]$. Therefore, we set $\beta = 0.2$ is in our implementation.

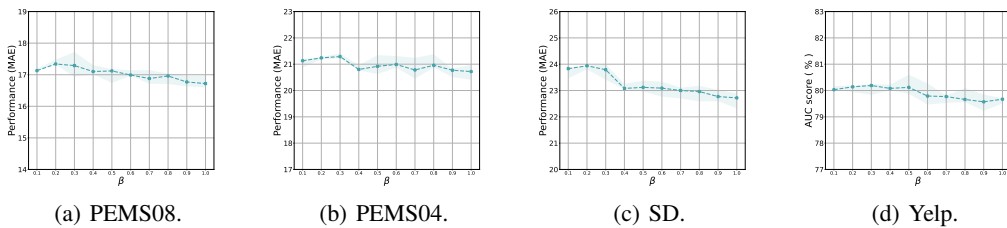

|                | (a) PEMS08. | (b) PEMS04. | (c) SD. | (d) Yelp. |

Figure 8: Sensitivity analysis of the hyperparameter $\beta$ on four real-world datasets.

We then analyze the sensitivity of hyperparameter $L$ in Eq. 12. As shown in Tab. 8, we study the performance of EpoD when $L$ is set to $[1, 10]$. We observe that increasing the value of $L$ tends to improve its performance. However, it is important to note that this improvement comes at the cost of increased time consumption. Therefore, we set $L = 5$ as a trade-off between performance and time consumption.

Table 8: The performance (MAE) of different $L$ in Eq. 12 on PEMS08.

| Model | 1 | 2 | 3 | 4 | 5 | 6 | 7 | 8 | 9 | 10 |
|---|---|---|---|---|---|---|---|---|---|---|
| EopD | 18.87 | 18.21 | 17.76 | 17.25 | 17.43 | 17.13 | 16.92 | 16.65 | 16.58 | 16.96 |

## G.2    Backbone Sensitivity Analysis

Our EpoD essentially provides a solution for temporal distribution shits issue in dynamic graphs. The most prominent characteristic of EpoD is the pluggability, which denotes that we can be applied to numerous existing backbones. In the task of traffic flow prediction, we explore the performance associated with the selection of different models as backbones as shown in Tab. 9. We get the following two **obs**ervations.

**Obs 1.** The EpoD-enhanced version consistently shows a significant improvement over the raw backbone. On large-scale datasets with distributional shits, EpoD tends to exhibit more substan-

Table 9: The MAE performance of EpoD with three different backbones ($12 \rightarrow 24$).

| MODEL | PEMS08 | PEMS04 | SD(2019-2020) | GBA(2019-2020) |
|---|---|---|---|---|
| ASTGCN [13] | 19.34 | 22.89 | 28.36 | 32.58 |
| EPOD+ASTGCN | 17.75 | 21.78 | 26.72 | 28.76 |
| DSTAGNN [19] | 17.56 | 21.22 | 26.34 | 30.11 |
| EPOD+DSTAGNN | 17.21 | **20.76** | 24.89 | 27.89 |
| AGCRN [3] | 17.30 | 21.19 | 26.19 | 28.74 |
| EPOD+AGCRN | **16.92** | 21.12 | **23.58** | **27.26** |

tial performance improvements. This highlights the effectiveness of our EpoD in addressing the distribution shift issue.

**Obs 2.** The expressive capacity of the backbone directly influences the predictive ability of the model enhanced by EpoD. Since the performance of AGCRN is already excellent, the enhancements introduced by EpoD often result in optimal results. This underscores the importance of selecting a proficient backbone model for forecasting.

### G.3 Interpretable Dynamic Subgraphs

In this subsection, we explore the interpretability of dynamic subgraphs under real-world scenarios. In recent years, the most notable temporal-distribution shift phenomenon is the outbreak of COVID-19. The introduction of LargeST [24] provides a chance for us to study this distribution shift of traffic flow under COVID-19. We choose a local sensors network in GBA dataset and reconstruct its evolution data at monthly intervals from 2019 ($t = 1, ..., T$) to 2020 ($t = T + 1, ..., T + k$). Then, we apply our EpoD trained on 2019 data to partition each snapshot into a bag of subgraphs, as shown in the top panel of Fig. 9. It is evident that the nodes tend to suppress communication among themselves when the distribution is shifting, and this phenomenon is particularly pronounced around the three blue nodes. The bottom panel of Fig. 9 visualizes the ground-truth of this sequence, which aligns with the information reflected in our dynamic subgraphs. This suggests that our EpoD exhibit sensitivity to environment changes.

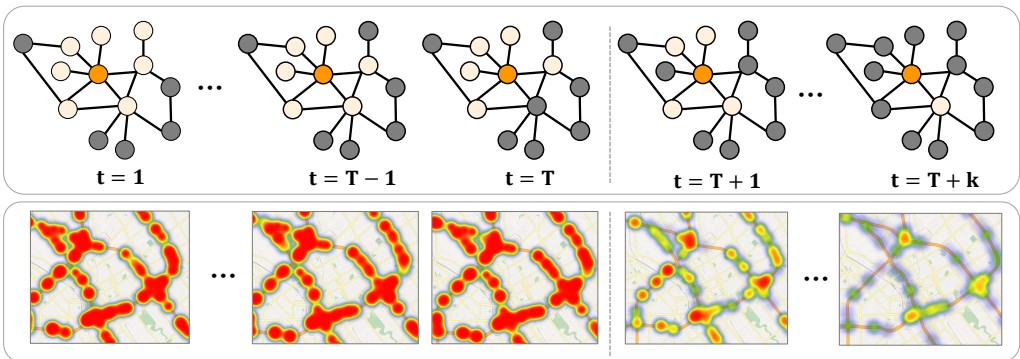

Figure 9: The interpretability of dynamic subgraphs within real-world scenarios.

