# OpenReview forum: "Improving Generalization of Dynamic Graph Learning via Environment Prompt"
_NeurIPS.cc/2024/Conference — NeurIPS 2024 poster_

### Official Review · Reviewer_YNjr · 2024-06-28

**Soundness:** 3
**Presentation:** 3
**Contribution:** 4
**Rating:** 7
**Confidence:** 4

**Summary:**

This work investigates the issue of spatio-temporal data distribution shift, which is a long-standing challenge in dynamic graph learning. First, the authors systematically analyze the limitations of existing works over OoD  challenge, and then propose a comprehensive solution to address their limitations.  Specifically, a self-prompted learning mechanism is proposed to extract underlying environment variables that potentially influence data distribution, and a novel causal pathway that leverages dynamic subgraphs as mediating variables is further introduced to effectively utilize the inferred environment embedding.  Comprehensive experiments on seven real-world datasets demonstrate the superiority of proposed EpoD.

**Strengths:**

- The motivation of this work is clearly described and convincing. This work proposes a novel framework to systematically tackle the  environment inference and utilization problem via profound understanding of the environment perception and limitations in existing OoD literatures.

- This work proposes some interesting and pioneering learning components for dynamic graphs, including self-prompted learning and dynamic subgraph learning.

- Sufficient experimental studies support the insights and framework of the paper. A toy dataset  verifies that the model can capture interpretable causal associations.

**Weaknesses:**

- It is widely agreed that unobservable environmental factors are the primary cause of shifts in data distribution. How does the self-prompted design ensure that prompt answers can capture unobserved environmental factors, without overlapping with the observable information?

- There are some  minor typos, e.g.,  in Line105,  more consistent with -> is more consistent with.

**Questions:**

-  How can we understand the sentence "Historical observations $X$ can be divided into the accessible environment features $X_X$ and observed labels with evolution patterns $Y_X$" in Line 90?

- This work presents a winding causal path to guide the utilization of environment variables. Is this causal path specific to dynamic graphs, can it be extended to other types of data?

- It is intuitively meaningful to use the asymmetric quantization principle to extract the node-centered subgraph. Is there more illustrative exampls to depict such asymmetry?

**Limitations:**

This manuscript contains sections "Broader impacts" and "Limitations and Future Works". The authors have described the impact of this work, and discuss its limitations in those sections.

---

> ### Author Rebuttal · Authors · 2024-08-07
>
> Dear Reviewer YNjr,
>
> We would greatly appreciate your positive comments on our work. We have carefully considered your questions and have provided detailed answers as follows.
>
> **W1. The design of our self-prompted mechanism.**
>
> There are some readily accessible environment factors in the original spatio-temporal data, such as recorded weather or the day of the week. These available environment variables are clearly not what our environment prompter intended to learn. Actually, our self-prompted mechanism is designed to avoid redundant learning, ensuring that our environment answers can  capture underlying environments that genuinely influences spatiotemporal evolution.  Specifically, the first term ${\rm{KL}}(\mathbb{P}_ \theta({\textbf{Z}_  E})||\mathbb{P}_ \phi(\textbf{Z}))$ in the learning objective of Eq. 5 constrains our environment prompts to be dissimilar with the representations of the original data.
>
>
> $$\mathop  {\min }\limits_ { \phi ,\theta, \textbf{P}}{{\cal L}_ P} =  \beta \mathbb{E}[ {\rm{KL}}(\mathbb{P}_ \theta({\textbf{Z}_ E})||\mathbb{P}_ \phi(\textbf{Z}))]- \mathbb{E}[ \log  {\mathbb{P}_ {\phi ,\theta}}({{\textbf{Y}_ \textbf{X}}}|{{\textbf{Z}_ E}})]$$
>
>
>
> **W2. Minor typo.**
>
> Thank you for your careful review on our manuscript. We have corrected the typo you pointed out and carefully reviewed our manuscript for other additional typos.
>
>
> **Q1. Interpretation of historical observations division.**
>
>
> As above described, the original spatio-temporal data may contain some accessible environment features that are crucial for inferring future spatio-temporal evolution patterns. We take a more fine-granular approach by dividing the historical observations $X$ into accessible features $X_X$, such as the day of the week, and the evolution of labels in the historical observations $Y_X$.
>
> Our design differs from existing autoregressive methods that focus solely on historical evolution $Y_X$, as well as from approaches that mix $X_X$ and $Y_X$ into a single modeling entity $X$. The advantage of our approach is that it provides a comprehensive understanding of spatio-temporal data from a data generation perspective and helps us infer underlying environmental variables (features).
>
>
>
>
> **Q2. Explanation of the winding causal path.**
>
>
> One of the most remarkable characteristics of spatio-temporal data is its dynamic nature, which results in the winding causal paths driven by environment shifts during evolution. In contrast, such winding causal paths are difficult to reasonably explain in non-temporal static data.
>
>
> There exists a concept of environment in the graph data, e.g. graph size, which is the main factor affecting the data distribution shift. However, the causal paths in the graph data are generally invariant, such as the decisive role of specific functional groups in a molecule on its properties. Therefore, changes in the environment are less likely to alter these causal paths. In contrast, dynamic graphs are far more complex. It is challenging to distill a stable and invariant causal path of spatio-temporal evolution. In conclusion, we contend that this winding causal path is unique to dynamic graphs and time series.
>
>
>
>
> **Q3. Asymmetric quantization principle.**
>
> Here we offer a more intuitive explanation to clarify the concept  of asymmetry.
>
>
> - In traffic networks, traffic flow always maintains a directional and asymmetric transmission between nodes. Asymmetry means that the number of vehicles traveling between nodes is not equal. When the environment changes, the original asymmetric transmission mode will alter. For example, the change of weather conditions always leads to a shift in the direction of traffic flow, and build new asymmetric transmission.
>
>
> - In social networks, the influence of individuals is asymmetrical. People of high social status always have a profound influence on their followers, but the reverse is not necessarily true.  When the environment changes, the original asymmetric relationship may shift or break, leading to the formation of a new asymmetric relationship.
>
> Inspired by such characteristics of dynamic graphs, we propose an asymmetric criterion to quantify environment effects. Further, we provides more interpretable insights through node-centered subgraphs.
>
>
> We greatly appreciate your constructive feedback on our work. We will carefully refine our manuscript based on your suggestions and look forward to your further comments!

---

### Official Review · Reviewer_5ZgN · 2024-07-08

**Soundness:** 3
**Presentation:** 4
**Contribution:** 3
**Rating:** 7
**Confidence:** 4

**Summary:**

This paper provides a novel dynamic graph learning framework EpoD to tackle the temporal distribution shift issue by exploiting prompt learning. EpoD addresses two limitations of existing works regarding inference and exploitation of unseen environments. The EpoD includes two modules, i.e., self-prompted environment inference component and dynamic subgraph learning component, where the former extracts underlying environment variables that potentially influence data distribution, while the latter effectively and interpretably exploits the inferred environment embedding for dynamic subgraph generation. The experiments design several distribution scenarios for evaluation and empirically demonstrate impressive performance under distribution shift scenarios.

**Strengths:**

**S1.**  Well-written and clearly organized. This work decouples the OOD issue into environment inference and utilization challenges, addressing each separately. Moreover, empirical examples and theoretical analysis enhance the accessibility of the paper for readers.

**S2.** Qualified technical contributions. EpoD addresses two limitations of existing works regarding inference and exploitation of unseen environments with two well-designed modules, environment prompting and dynamic subgraph learning, which innovatively leverage successful experiences from LLMs to address the OOD generalization issue in dynamic graph learning.

**S3.** Good experiment designs and comprehensive evaluations. The authors especially design several distribution scenarios for OOD task evaluation with consideration of the COVID-19 period.  Seven cross-domain real-world dynamic graph datasets are selected to evaluate the performance of EpoD.

**Weaknesses:**

**W1.** More detailed explanations and analysis will be more welcome and help readers better understand this manuscript. What is the unique superiority of the self-prompted learning mechanism in inferring unseen environments in spatio-temporal graph data?

**W2.** It seems that self-prompt is borrowed from LLMs, thus what is the intuitive designing idea of self-prompt learning here, and how it works for environment inference?

**W3.** Any other considerations besides efficiency when designing node-specific and time-shared prompt tokens?

**W4.** There are many methods to extract subgraphs, the most popular one is to drop edges to realize the partition of graph data. What do you think are the advantages of node-centered dynamic subgraphs?

**W5.** Some typos:

- In line 254, "dynamics graph" should be modified to "dynamic graph".

- In line 385, "we" should be modified to "our".

**Questions:**

Please answer the questions in Weaknesses.

**Limitations:**

The authors have discussed the broader social impacts and limitations of their work.

---

> ### Author Rebuttal · Authors · 2024-08-07
>
> Dear Reviewer 5ZgN,
>
> Thank you for taking the time to review our work. We greatly appreciate your positive feedback and will address your comments with careful consideration.
>
>
> **W1. Advantages of self-prompted learning mechanism.**
>
>
> Our self-prompted mechanism has two key advantages that distinguishes from existing works:
>
> 1. Our self-prompted mechanism focuses on modeling historical observations to infer the environmental space, which has not been explored. This approach ensures that the extracted environmental space aligns with spatio-temporal evolution patterns, rather than being an arbitrary expansion.
>
> 2. We do not assume a predefined environmental scale. Our design infers unseen environment factors into a continuous space, which is remarkably distinctive from previous methods with predefined and discrete environment scales.
>
> Actually, the above analysis has already been discussed in Sec. 3.1.
>
>
> **W2. Explanations of environment inference process.**
>
>
> Our self-prompted learning framework achieves the inference of unseen environment by a novel squeezing strategy, which is derived from analyzing the relationships between environmental variables $\textbf{E}$, observable features $\textbf{X}_ \textbf{X}$, and evolutionary patterns $\textbf{Y}_ \textbf{X}$. Although inspired by LLMs, this insight makes our self-prompted module distinct from LLMs in both its objectives and implementation manners.
>
> Specifically, given the availability of $\textbf{X}_ \textbf{X}$,  $\textbf{Y}_ \textbf{X}$ and $\textbf{C}$, we adopt the strategy that infers  the environment in historical observations, i.e., $\textbf{E} \leftarrow {g_ \theta }(\textbf{X}_ \textbf{X}, \textbf{Y}_ \textbf{X},\textbf{C})$.
>
> $$\mathbb{P}(\textbf{X}, \textbf{Y}|\textbf{E},\textbf{C}) = \mathbb{P}(\textbf{Y}|\textbf{X},\textbf{E},\textbf{C})\mathbb{P}(\textbf{X}|\textbf{E},\textbf{C})$$
>
>
>
>
> **W3. Design principles of learnable prompt tokens.**
>
> Our prompt design is based on the principle that nodes in a dynamic graph have a baseline environment, but as time evolves, the environment tends to deviate from this baseline environment, resulting in data distribution shifts.
>
> The node-wise and temporal-shared environment prompts $P$ we designed effectively construct such the baseline environments for each node, using learnable parameters to capture such environment factors in spatio-temporal data. Furthermore, the prompt answers, obtained through the interactive prompt-answer squeezing mechanism, reflect  the real environment representation at each time step.
>
>
> **W4. The advantages of node-centered dynamic subgraphs.**
>
> The primary advantage of node-centered subgraph is its ability to accurately reflect the real-world influence of environment factors on dynamic graphs. Concretely, this design stems from a profound understanding of dynamic graphs, which is the shifts of environments  invariably lead to changes in the asymmetric correlations between nodes. For example, in a traffic network, environmental changes are often reflected by shifts in the flow direction between nodes.
>
> Our node-centered dynamic subgraph extractor can capture such node-specific asymmetry, with each node having a unique subgraph tailored for its environmental state. Compared to subgraphs obtained by simply dropping edges, node-centered subgraphs have a superior ability to capture and characterize spatio-temporal distribution shifts.
>
>
> **W5. Some typos.**
>
> Thank you very much for your careful review, we will carefully check our manuscript to correct all typos.
>
> We are very grateful for the constructive comments you provided on our work. Next, we will carefully refine our manuscript based on your suggestions. Looking forward to your further feedback!

---

> > ### Comment · Reviewer_5ZgN · 2024-08-08
> >
> > The authors have addressed most of my concerns. I suggest acceptance after careful refinement of the manuscript.

---

> ### Author Response · Authors · 2024-08-08
> **Further feedback for 5ZgN**
>
> Dear Reviewer 5ZgN,
>
> We would like to express our deep gratitude for your professional comments, which will greatly enhance our work. We are committed to refining our manuscript by offering more intuitive explanations, and conducting a thorough discussion of related techniques.
>
> Thank you once again, and we hope you have a wonderful day!
>
> Authors of Paper 4911.

---

### Official Review · Reviewer_XPEX · 2024-07-10

**Soundness:** 3
**Presentation:** 2
**Contribution:** 3
**Rating:** 6
**Confidence:** 3

**Summary:**

The paper is about dynamic graph learning, which is an interesting topic. The authors propose a novel dynamic graph learning model named EpoD based on prompt learning and structural causal model to comprehensively enhance both environment inference and utilization. The paper is well written and well organized. However, there are several concerns in the current version of the paper that addressing them will increase the quality of this paper.

**Strengths:**

1 New perspectives on theory.

2 Good writing.

3 Fully experimented.

**Weaknesses:**

1 Regarding the OOD research of dynamic graphs, I have a question: when new nodes appear in the data (rather than the known number of nodes), how should the model construction be handled?

2 Would like to see a discussion of computational complexity, both time complexity and space complexity.

3 The details of the datasets should be included in the text (rather than in the appendix), such as the size of the datasets. In addition, it seems that the size of these datasets is limited.

**Questions:**

As above.

**Limitations:**

Yes.

---

> ### Author Rebuttal · Authors · 2024-08-07
>
> Dear Reviewer XPEX,
>
> Thank you very much for your thoughtful review! We have carefully considered your comments, and we will provide detailed responses below. We hope these details can address your concerns.
>
> **W1. The ability to counter structural distribution shifts of EpoD.**
>
> Addressing node-scale distribution shifts is a key challenge in spatio-temporal OOD research. Actually, our EpoD can effectively resist structural distribution shifts with simple modifications. Here, we introduce a strategy of modifying EpoD to counter these shifts, along with the rationale behind it.
>
> - The learnable prompt tokens $\mathbf{P} \in \mathbb{R}^{N \times d}$ is the only component in EpoD sensitive to node scale. To counter structural distribution shifts, we can introduce a neighbor-sharing prompt strategy. When a new node is added, the prompt token for this new node will be populated based on the prompt token of the node to which it is most closely connected, resulting in a new environment prompts $\mathbf{P} \in \mathbb{R}^{(N + 1) \times d}$. When a node is removed, we eliminate the corresponding node's prompt token, resulting in a new environment prompt $\mathbf{P} \in \mathbb{R}^{(N - 1) \times d}$. The new prompt is utilized to update the following learning flow.
>
> - The insight of this design lies in that both temporal and structural distribution shifts can be distilled into the shifts of environmental variables. Specifically, temporal distribution shifts arise from the shift of environmental factors, while structural distribution shifts essentially impact the expression of environments in spatiotemporal data.
>
> To verify the effectiveness of this modification, we conduct some preliminary empirical studies on the adjusted SD-2019 dataset, i.e., SD-2019-MaskNodes. Specifically, we randomly mask 10\% of the nodes in the training set and restore them to 716 nodes in the test set. We present the final performance (MAE) by averaging the results from two runs conducted on an NVIDIA A100-PCIE-40GB. Since none of the existing methods are designed to address the structure distribution shift, we were unable to find a suitable baselines for comparison. As shown in the following table, EpoD shows only a slight performance degradation on SD-2019-MaskNodes, and outperforms most methods on SD-2019.
>
> |      | SD-2019 | SD-2019-MaskNodes|
> |   :--:   | :--:   | :--:   |
> | EpoD(MAE) | 16.89 | 17.12 |
>
>
> **W2-1. Time complexity analysis.**
>
> We analyze the efficiency of EpoD theoretically and practically.
>
> - We utilize $|V|$ and $|E|$ to denote the number of nodes and edges in the graph, $d$ to represent the dimension of implicit representation, and $T$ to represent the time step of historical observations. The time consumption mainly comprises three components: the spatio-temporal graph aggregation process, the prompt answer process, and the dynamic subgraphs sampling process. The time complexity of the spatio-temporal aggregation is $O(T \cdot (|E| \cdot d + |V| \cdot d^2))$. The prompt answer process primarily involves a cross-attention operation, with a time complexity of $O(T \cdot |V| \cdot d)$. The dynamic subgraphs sampling module implements node-centered sampling, with a time complexity of $O(T \cdot |V|)$. Therefore, the time complexity of EpoD is $O(T \cdot d  \cdot |E| + T \cdot (1+d+d^2) \cdot |V|)$. In conclusion , EpoD exhibits linear time complexity concerning the number of nodes and edges, which is competitive with existing dynamic GNNs such as DIDA, EAGLE, and CaST.
>
> - We also conduct efficiency comparisons of EpoD, DIDA, and EAGLE in COLLAB, Yelp, and ACT datasets, measuring the time taken per epoch (s/epoch). All experiments are run on an NVIDIA A100-PCIE-40GB. Empirically, we observe that the operational efficiency of our method is competitive with existing approaches.
>
> |      | DIDA | EAGLE| EpoD |
> |   :--:   | :--:   | :--:   |  :--: |
> | COLLAB | 11.21 | 12.05 |  11.84 |
> | Yelp |  6.89  | 7.38 |  7.34  |
> | ACT | 9.27 |  9.76  |  9.59 |
>
>
>
> **W2-2. Parameter scale analysis.**
>
> The increased complexity of EpoD is mainly reflected in the introduction of the environment prompt module. As shown in following Table, we compare the number of parameters with several baseline models on SD-2019. We find that the increase of our model size was not prominent. Given the performance  achieving 1.8\% improvements on MAE  under temporal shifts, such sacrifice in parameter complexity is acceptable and deservable.
>
> |      | GWNET | STGNCDE| DSTAGNN | CaST| EpoD |
> |   :--:   | :--:   | :--:   |  :--: |:--:   | :--:   |
> | Parameters | 311K | 729K |  3.9M | 652K |  894K |
>
>
>
> **W3. The details of the datasets.**
>
> Due to the page limitations of each submission, we had to place some static content, including the dataset description and baseline details, into the appendix. We appreciate your feedback and will move Tab. 5 into the main text in the next version.
>
> Moreover, we have already chosen some of the largest datasets in the field, such as the large-scale traffic datasets LargeST (SD, GBA), as well as large-scale social network datasets COLLAB and ACT. When new large-scale datasets are introduced, we will promptly validate our EpoD on them.
>
>
> Thank you again for your constructive comments on our work. We will take your comments into account to further improve our manuscript. Looking forward to your feedback!

---

### Official Review · Reviewer_Zz1E · 2024-07-11

**Soundness:** 3
**Presentation:** 3
**Contribution:** 3
**Rating:** 7
**Confidence:** 4

**Summary:**

The paper introduces EpoD, a novel dynamic graph learning model that leverages prompt learning and structural causal models to address out-of-distribution (OOD) generalization challenges. EpoD features a self-prompted learning mechanism for inferring environment variables and a node-centered subgraph extractor to capture data distribution shifts. Extensive experiments across various real-world datasets demonstrate EpoD's superior performance and interpretability, highlighting its potential for practical applications in fields like traffic forecasting and social network analysis.

**Strengths:**

1. The paper introduces a novel dynamic graph learning model named EpoD, leveraging prompt learning and structural causal models to address environment inference and utilization. The self-prompted learning mechanism and node-centered subgraph extraction represent significant advancements in the field.

2. The authors provide a solid theoretical framework supporting the generalizability and interpretability of EpoD. The incorporation of structural causal models and dynamic subgraphs as mediating variables showcases a deep understanding of the underlying principles of dynamic graph learning.

3. The paper presents extensive experiments across seven real-world datasets from diverse domains, demonstrating the superiority of EpoD over several baseline models. The inclusion of a toy example experiment further validates the interpretability and rationality of the proposed model.

4. The results indicate that EpoD consistently outperforms existing methods in terms of both mean absolute error (MAE) and root mean square error (RMSE) for traffic flow prediction, as well as AUC scores for social link prediction tasks. The significant improvements on large-scale datasets highlight the model's robustness and scalability.

5. The dynamic subgraph extraction process and the use of node-centered subgraphs enhance the interpretability of the model. The paper effectively demonstrates how dynamic subgraphs can capture the impact of environment variable shifts on data distribution.

6. The model's ability to generalize across different domains and datasets suggests strong potential for practical applications in various fields, such as traffic forecasting, social network analysis, and air quality prediction.

**Weaknesses:**

To be honest, I did not find obvious drawbacks of this work. Some minor revision might be helpful:

- use the mentioned all-in-one (citation [39] in their manuscript), a more flexible graph prompt as your environment vector. Or see the impact of different graph prompt formats.
- The discussion on future work and the limitations of the current approach could be expanded. For example, I think this work might be helpful to be applied in some sociological analysis. The environment concept might be also inspiring to the similar concept in this paper: Self-supervised Hypergraph Representation Learning for Sociological Analysis. TKDE. 2023.


Overall, this paper presents a significant contribution to the field of dynamic graph learning, with innovative methods and strong experimental results.

**Questions:**

N/A

**Limitations:**

See in the weakness

---

> ### Author Rebuttal · Authors · 2024-08-07
>
> Dear Reviewer Zz1E,
>
> Thank you for your valuable time in reviewing our manuscript. We greatly appreciate your positive comments on our work, and your insights are invaluable to us. We will provide detailed replies to your questions next.
>
> **W1. More extensive prompts design.**
>
> One of the purposes of our work is to introduce the concept of prompt learning for the preliminary exploration of spatio-temporal OOD. This is why we designed a relatively simple node-wise learnable vector prompts. Excitingly, the effectiveness of EpoD demonstrates that prompt-based methods can bring new vitality and potential to spatio-temporal graph learning. Therefore, as you suggested, exploring more diverse forms of prompt $\mathbf{P}$ tailored for spatio-temporal scenarios can be the next direction pursue. Specifically, we outline the following analysis and plans for future research.
>
> Ref.[1] verified that graph-structured prompts can enrich the semantics of graph learning. However, directly applying this method to spatio-temporal graph learning may not offer strong interpretability. The reason lies that the dynamic nature of spatio-temporal graph introduces new challenges, particularly concerning the impact of temporal heterogeneity on the inserting patterns and number of designed prompts.
>
> - Regarding the graph-structured prompts $\mathbf{P}$, we will explore whether to insert $\mathbf{P}$ into $G$ in a manner similar to [1], or to design an interactive framework to achieve semantic learning as proposed in EpoD.
>
> - Further, we will investigate whether it is necessary to design temporal-specific graph prompts to fully capture the underlying environmental information in each time step.
>
>
>
> **W2. Sociological analysis.**
>
> With a preliminary investigation,  we find that social networks share the similar spatiotemporal evolution process of the ST graph investigated in this work. To this end, our work has significant potentials for application in sociological analysis, leading to a promising direction of our future research. As Ref.[2] mentioned, regarding the social equivalence and social conformity effect, individuals in social networks tend to either form their own groups or become assimilated into the environment over time. We argue that the key characteristics of this process are its gradual temporal progression and environmental orientation.
>
> On one hand, the changes of individuals are usually not sudden; instead, they gradually become apparent over time and eventually lead to transformation. This temporal progression provides a solid foundation for studying social networks from a spatio-temporal perspective.
>
> On the other hand, individual evolution typically aligns with the direction of the environment. However, the environmental space within social networks is complex and challenging to capture. As a result, inferring the local or global social environment within these networks becomes a crucial challenge. Fortunately, the environment inference and subgraph learning in our EpoD have the potential to effectively address this challenge.
>
> These two aspects indicate that our work has the potential to be applied to sociological analysis. Specifically, the environment inference and subgraph learning mechanism we proposed naturally align with these challenges. We are very interested in further exploring this direction in our future work, and we will incorporate above analysis into our manuscript.
>
> Thank you for your insightful comments, many of which deepen our understanding on this work. They also inspire our future research directions. We appreciate your thorough review once again.
>
>
> **References:**
>
> [1] All in one: Multi-task prompting for graph neural networks.
>
> [2] Self-supervised Hypergraph Representation Learning for Sociological Analysis.

---

> > ### Comment · Reviewer_Zz1E · 2024-08-08
> >
> > I have carefully read your response. Thanks

---

> ### Author Response · Authors · 2024-08-08
> **Further feedback for Zz1E**
>
> Dear Reviewer Zz1E,
>
> Thanks again for your professional insight, which will significantly enhance our work.  If you have any questions, please don't hesitate to let me know. We will answer your concerns carefully. We truly appreciate your support and wish you a wonderful day!
>
> Authors of Paper 4911.

---

### Decision · Program_Chairs · 2024-09-25

**Decision:**

Accept (poster)

**Comment:**

This paper presents a new dynamic graph learning method, which leverages prompt learning and structural causal models to tackle the challenge of out-of-distribution generalization. Reviewers raised some comments regarding technical details, complexity, assumptions, and paper writing, which have been mostly addressed in the authors' rebuttal. The authors are highly encouraged to improve the final version of their paper according to the suggestions from reviewers.